# PADS-TAL: Padding-Annealed Diffusion Sampling in Text-Aware Latent Space for Robust and Diverse Text-to-Music Generation

Taekoan Yoo [* 1]  Wonkyung Jung [* 1]  Kyunghun Kim [* 1]  Kyeongbo Kong [2]

## Abstract

Text-to-Music diffusion models are increasingly used in real-world applications, yet deployment remains challenging: generations can collapse to limited patterns even with diverse initial noise and prompts, and inference-time diversity control often harms text alignment and fidelity by distorting key prompt cues established in early denoising. To address this, we propose *Padding-Annealed Diffusion Sampling*, which perturbs only a padding-indexed subspace while keeping non-padding conditioning fixed, enabling controlled exploration with reduced semantic drift. However, in a text-unaware VAE latent space, such exploration is less likely to stay within genre-faithful neighborhoods, limiting genre-consistent diversity. We therefore introduce *Text-Aware Latent space* that aligns local neighborhoods with text-implied genre structure, promoting genre-consistent diversity. Together, the two techniques form a unified pipeline that, compared to prior methods that perturb the full conditioning, achieves a better text alignment–diversity trade-off: at comparable text alignment, it delivers 15.4% higher diversity with a relatively small fidelity drop, and further improves within-genre diversity by 71.6%. Generated samples are available at https://pads-tal.github.io/PADS-TAL.io

## 1. Introduction

Over the past few years, Text-to-Music (T2M) generation has progressed along two main paradigms: transformer-

---

*Equal contribution – Taekoan Yoo led the project, with substantial contributions from Kyunghun Kim and Wonkyung Jung. [1]AI Tech Lab., NHN Corp., Seongnam-si, Gyeonggi-do, Republic of Korea [2]Pusan National University, Busan, Republic of Korea. Correspondence to: Kyeongbo Kong <kbkong@pusan.ac.kr>.

*Proceedings of the $43^{rd}$ International Conference on Machine Learning*, Seoul, South Korea. PMLR 306, 2026. Copyright 2026 by the author(s).

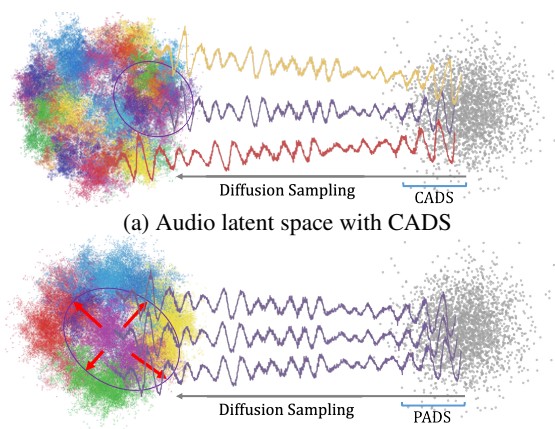

(a) Audio latent space with CADS

(b) Text-aware latent space with PADS (Ours)

*Figure 1.* Conceptual comparison between the standard diffusion model with CADS and ours. Shown for illustration only.

based autoregressive models (Copet et al., 2023; Agostinelli et al., 2023; Lei et al., 2025; Liu et al., 2025a; Yuan et al., 2026; Dhariwal et al., 2020) and diffusion-based models. While early diffusion T2M systems primarily relied on U-Net-style architectures (Schneider et al., 2024; Chen et al., 2024; Forsgren & Martiros, 2022; Lam et al., 2023; Zhu et al., 2023; Huang et al., 2023c; Li et al., 2024a; Huang et al., 2023b), recent work has increasingly adopted Diffusion Transformers (DiTs), which combine diffusion modeling with Transformer backbones that scale well for large generative modeling (Evans et al., 2025; 2024a;b; Novack et al., 2025; Fei et al., 2024; Bai et al., 2024). A representative example is Stable Audio Open (SAO) (Evans et al., 2025), which employs a DiT architecture in a Variational Autoencoder (VAE)-compressed latent space and is widely used as a reproducible reference baseline in prior work.

To bring T2M generation to practical deployment, it is crucial to acquire and curate large-scale music data that are both diverse and properly licensed. In practice, however, data collection and curation are expensive and licensing constraints remain substantial, often limiting coverage across melodic patterns and stylistic variations. As a result, text-conditioned diffusion models (DMs) can exhibit limited diversity, producing outputs that degenerate into repetitive patterns even when varying the initial noise or the prompt condition. Importantly, the diversity users typically expect in T2M is not merely unconditional variation across sam-

ples, but variation that preserves the attributes specified in the prompt, including global ones such as genre. Despite its practical importance, systematic methodologies for controlling and improving this form of diversity remain relatively underdeveloped.

In the Text-to-Image (T2I) literature, numerous inference-time techniques have been proposed to increase sample diversity without additional training (Sadat et al., 2024; Corso et al., 2024; Wu et al., 2025). A representative example is Condition-Annealed Diffusion Sampling (CADS) (Sadat et al., 2024), which injects Gaussian noise into the conditioning embeddings during denoising, with stronger perturbations at early timesteps. This inference-only strategy is attractive in practice because it can expand diversity without changing model parameters, by enabling sampling to explore multiple plausible realizations around a fixed condition. Given its inference-only nature, CADS-style conditioning perturbation is a natural starting point for improving diversity in T2M, as it introduces controlled stochasticity in the conditioning signal and can, in principle, generalize across various prompts. However, we find that naively transferring such perturbations to T2M can be brittle: as the perturbation strength increases, it can markedly degrade text–audio alignment and audio fidelity, thereby undermining *text-aligned diversity*. Moreover, improvements in *text-aligned diversity* do not necessarily translate into higher *genre-consistent diversity*: stronger perturbations may broaden sampling exploration and occasionally increase genre-wise coverage; however, such gains are typically small or erratic, and thus still often lead to outputs that fail to exhibit the requested genre-specific characteristics.

We argue that this phenomenon is closely related to diffusion sampling dynamics and the characteristics of the T2M domain. Prior work (Ho et al., 2020) has observed that diffusion generation tends to establish a coarse global scaffold—which is particularly crucial in music, where prompts often emphasize global attributes such as genre, mood, and overall structure—during early denoising stages. Accordingly, strong perturbations to conditioning embeddings in the early high-noise regime can distort the signals that govern the overall piece, making it easier for the generation trajectory to drift toward regions that are only weakly related to the intended condition (details in Appendix G.3). This effect can be further exacerbated when the sampling latent space is not well aligned with dataset semantics, such that nearby regions in latent space may not correspond to genre-faithful variations. These observations motivate our unified approach (Fig. 1). We provide supporting evidence through controlled ablations of early-stage perturbations (Fig. 2) and latent-space neighborhood analyses of genre structure (Appendix D).

In this paper, we propose two complementary solution axes

that follow directly from the above analysis: (i) restricting where conditioning perturbations are applied at inference time, and (ii) aligning the sampling representation so that local neighborhoods better reflect text semantics. First, we introduce *Padding-Annealed Diffusion Sampling* (PADS), an inference-only sampling strategy that stabilizes conditioning perturbations. In an embedded text-conditioning sequence, explicit semantics are carried by the non-padding positions. We therefore keep all non-padding embeddings fixed and instead inject timestep-dependent Gaussian noise into a low-salience padding-indexed subspace at early steps to encourage exploration while minimizing interference with semantic cues. Second, we introduce *Text-Aware Latent* (TAL) space to improve genre-consistent generation at the representation level. Latent Diffusion Models (LDM) (Rombach et al., 2022) learn diffusion in a VAE latent space rather than raw signal space, yet the VAE in conventional pipelines is typically trained for audio reconstruction without text. Consequently, the latent-space neighborhood structure need not align with text semantics—particularly for global attributes such as genre. We therefore redesign the VAE to be text-aware by jointly leveraging text and audio representations, encouraging semantically similar audio–text pairs to become neighbors in latent space. As conceptualized in Fig. 1, training diffusion in TAL promotes sampling around global-consistent neighborhoods, providing a stronger foundation for genre-consistent generation under inference-time perturbations, in contrast to directly perturbing conditioning in a text-unaware audio latent space.

In summary, we propose a unified pipeline to improve text-aligned diversity while preserving genre consistency in diffusion-based T2M models.

- We propose PADS, an inference-time strategy that restricts conditioning perturbations to a low-salience subspace to expand exploration while minimizing corruption of semantic cues.
- We propose TAL, a text-aware learned latent space that encourages the DM's representation to better reflect global text semantics, such as genre.
- On our benchmarks, our unified pipeline improves diversity and genre coverage under matched text alignment, compared to prior full-conditioning perturbation in DiT-based T2M.

**Conflict of Interest Disclosure.** The authors declare no financial conflicts of interest. This work does not evaluate or promote any company product or commercially used model.

## 2. Related Work

### 2.1. Diffusion Models for Text-to-Music Generation

Recent progress in T2M generation has been propelled by DMs, which enable high-quality synthesis with improved

structural coherence. As an early study, Riffusion (Forsgren & Martiros, 2022) fine-tuned Stable Diffusion (Rombach et al., 2022) in the spectrogram domain, demonstrating that DMs originally developed for image generation can be adapted to music generation. Subsequent works (Schneider et al., 2024; Li et al., 2024a) have further explored architectural designs that better preserve consistency in longform generation while modeling complex musical structure. More recently, latent-space diffusion frameworks coupled with transformer backbones have become a widely adopted paradigm due to their reproducibility and strong performance (Evans et al., 2025; Gong et al., 2025; Le Lan et al., 2024), SAO (Evans et al., 2025) serving as a representative open baseline in this line of work. However, because SAO uses a VAE latent space optimized primarily for audio reconstruction, its geometry may favor perceptual fidelity over higher-level semantic structure. Motivated by this property, we adopt SAO as our base model and reshape its latent representation to improve genre consistency.

## 2.2. Encouraging Exploration in Diffusion Sampling

A broad line of research studies how to steer diffusion sampling at inference time to shape the generated distribution. Some works improve coverage of underrepresented regions using separately trained classifiers (Sehwag et al., 2022; Um et al., 2024) or prompt optimization at inference time (Um & Ye, 2025). More structural approaches intervene directly in the sampling process, either through sampling-time perturbations such as OSCAR (Wu et al., 2025), auxiliary diversity guidance such as Particle Guidance (Corso et al., 2024), c-VSG (Askari Hemmat et al., 2024), and SPARKE (Jalali et al., 2025), or control of existing guidance as in LIG (Kynkäänniemi et al., 2024). In contrast, CADS (Sadat et al., 2024) promotes exploration by perturbing the conditioning information injected into the model, rather than modifying trajectories directly or adding an auxiliary diversity guidance term. However, since CADS perturbs the conditioning embedding as a whole, even small perturbations can lead to semantic drift or degraded fidelity in T2M. Motivated by this limitation, we propose PADS to secure sufficient exploration freedom without damaging the core semantics of the condition.

## 2.3. Mixture-of-Experts Multimodal VAEs

Mixture-of-Experts multimodal VAEs (MoE-mVAEs) aim to capture semantic factors shared across different modalities as a shared latent, while preserving modality-specific information as a private latent. The original MMVAE (Shi et al., 2019) treats each modality's encoder as an expert and aggregates unimodal posteriors using a Mixture-of-Experts formulation, enabling the model to learn the likelihood of other modalities via the shared latent even when only a single modality is provided. However, it has been noted that

concentrating information into a single shared latent can dilute modality-specific details. Along this line, mmJSD (Sutter et al., 2020) further supports the benefit of introducing modality-specific latent subspaces, while MMVAE+ (Palumbo et al., 2023) highlights the importance of properly balancing shared and private pathways. Inspired by this line of work, we adopt a text–audio MoE-style shared–private VAE to construct a text-aware latent space that preserves audio-specific information while incorporating shared text semantics, and then train a DM in the resulting latent space.

## 3. Method

In this section, we describe our unified pipeline. We first introduce PADS, which perturbs only a low-salience padding subspace while keeping non-padding conditioning embeddings unperturbed, and then introduce TAL, a text-aware latent space that aligns local neighborhoods with text semantics to support genre-consistent exploration; finally, we combine them and analyze their effects.

## 3.1. Padding-Annealed Diffusion Sampling

CADS (Sadat et al., 2024) injects annealed Gaussian noise into the full conditioning embedding $c \in \mathbb{R}^{L \times D}$ (sequence length $L$, feature dimension $D$) at timestep $t$ to mitigate overconcentration in conditional representations and promote broader sampling. Concretely, CADS corrupts $c$ according to an annealing schedule:

$$\hat{c} = \sqrt{\gamma(t)}\, c + s\sqrt{1 - \gamma(t)}\, n,$$

$$\text{where } \gamma(t) = \begin{cases} 1 & \text{if } 0 \leq t \leq \tau_1, \\ \frac{\tau_2 - t}{\tau_2 - \tau_1} & \text{if } \tau_1 < t < \tau_2, \\ 0 & \text{if } \tau_2 \leq t \leq 1, \end{cases} \quad (1)$$

where $s$ is the initial noise scale and $n \sim \mathcal{N}(\mathbf{0}, \mathbf{I})$ with $n \in \mathbb{R}^{L \times D}$. The cutoff thresholds $\tau_1$ and $\tau_2$ define the annealing schedule $\gamma(t)$.

While effective in some domains, uniformly perturbing all token embeddings can directly corrupt semantically critical cues, potentially causing unstable text alignment and degraded sample quality. Fig. 2a illustrates how alignment and fidelity vary with perturbation magnitude under classifier-free guidance (CFG) (Ho & Salimans, 2021) with $\omega_{\text{cfg}} = 7.0$. To compensate for reduced diversity under large guidance, CADS suggests stronger perturbations, using smaller $\tau_1$ and larger $s$, as $\omega_{\text{cfg}}$ increases. However, in T2M, we observe sharp drops in text alignment or fidelity even when $\omega_{\text{cfg}}$ is high and $\tau_1$ is not particularly small, indicating that such sensitivity to conditioning perturbations is not a minor issue in T2M.

To address this, we propose *Padding-Annealed Diffusion Sampling* (PADS), an inference-time strategy that can be

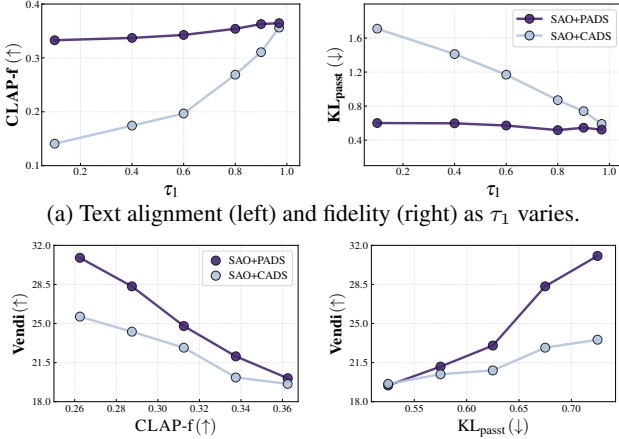

(a) Text alignment (left) and fidelity (right) as $\tau_1$ varies.

(b) Diversity at matched CLAP-f (left) or $\text{KL}_{\text{passt}}$ (right)

*Figure 2.* Comprehensive analysis of CADS and PADS under varying noise levels on SongDescriber.

viewed as a generalization of CADS via mask-controlled conditioning perturbations. We first partition the fixed-length conditioning sequence into a salient subspace and a low-salience subspace, $\boldsymbol{c} = [\boldsymbol{c}_{\text{sal}}; \boldsymbol{c}_{\text{low}}]$, where $\boldsymbol{c}_{\text{sal}} \in \mathbb{R}^{L_{\text{sal}} \times D}$ denotes the subspace most critical for preserving text semantics; we then inject annealed noise only into $\boldsymbol{c}_{\text{low}}$ to expand exploration while stabilizing text semantics. Optionally, to guarantee a minimum perturbation budget, we reserve the last $L_{\text{min}}$ positions as perturbation slots so that $L_{\text{min}}$ locations are always available for noise injection.

Formally, PADS constructs the perturbed conditioning embedding with a selective noise mask:

$$\hat{\boldsymbol{c}} = \boldsymbol{c} \odot (1-M) + \left(\sqrt{\gamma(t)}\,\boldsymbol{c} + s\sqrt{1-\gamma(t)}\,\boldsymbol{n}\right) \odot M, \quad (2)$$

where $M \in \{0,1\}^{L \times D}$ is a row-wise constant mask selecting low-salience positions. With indices $i \in \{1, \ldots, L\}$, we define:

$$M_i \in \{0,1\}^D = \begin{cases} \mathbf{1}_D, & \text{if } i > L_{\text{sal}} \text{ or } i > L - L_{\text{min}}, \\ \mathbf{0}_D, & \text{otherwise.} \end{cases} \quad (3)$$

This mask-based view makes PADS a mask-controlled generalization of CADS: CADS perturbs the full conditioning embedding ($M = \mathbf{1}_{L \times D}$), whereas PADS perturbs only a selected low-salience subspace. In T2M, PADS instantiates $\boldsymbol{c}_{\text{low}}$ as the padding-indexed subspace of the text-conditioning sequence and restrict conditioning noise to padding positions, which are expected to be weakly coupled to prompt semantics and thus preserve text alignment. Fig. 3 empirically supports the claim that this instantiation is alignment-safe: as perturbations are swept from the padding end toward semantic embeddings, alignment remains largely unchanged while the sweep stays within padding positions, but drops sharply once it reaches the semantic part.

In practice, many T2M systems accommodate variable-length prompts via zero-masked padding or attention-

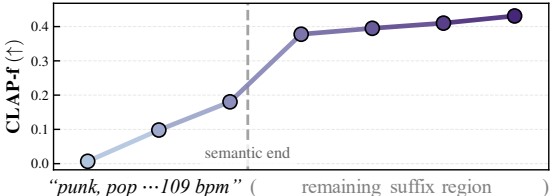

*Figure 3.* CLAP-f under a noise sweep from padding to semantic token embeddings.

masking mechanisms (see Appendix B.4); under these common designs, $\boldsymbol{c}_{\text{low}}$ is often treated as low-salience or approximately semantically inert, making it a natural target for controlled noising that avoids directly corrupting prompt semantics. At the same time, prior studies (Toker et al., 2025; Zhuang et al., 2024; Feng et al., 2023; Li et al., 2024b; Liu et al., 2025b) suggest that padding-related positions can still carry useful but minor signals. PADS is consistent with this view: it utilizes any information in $\boldsymbol{c}_{\text{low}}$ while applying annealed perturbations in a controlled manner, thereby increasing exploration without discarding—and potentially still benefiting from—any non-trivial signals that may reside in the suffix.

Consequently, for a fair diversity comparison, we report Vendi at matched CLAP-f or matched $\text{KL}_{passt}$ (Fig. 2b). Under both matching criteria, PADS achieves higher Vendi than CADS, suggesting that limiting perturbations away from semantically salient coordinates enables broader exploration without directly corrupting semantic cues.

**Intuition behind padding annealing.** Building on the Langevin-dynamics interpretation previously established for CADS—where adding Gaussian noise to the entire conditioning embedding induces an additional Langevin-like stochastic term in the latent update—we hypothesize that the same mechanism can be used to provide a principled restriction by injecting stochasticity only through the less influential part of the conditioning embedding, thereby promoting exploration without directly corrupting salient semantic information.

Starting from the conditional score $\nabla_{\boldsymbol{z}_t} \log p_t(\boldsymbol{z}_t \mid \hat{\boldsymbol{c}})$, we perturb only the low-salience component: $\hat{\boldsymbol{c}} = [\boldsymbol{c}_{\text{sal}}; \boldsymbol{c}_{\text{low}} + \sigma_c \boldsymbol{n}]$ with $\boldsymbol{n} \sim \mathcal{N}(\mathbf{0}, \mathbf{I})$. To connect this perturbation to the probability flow ODE update, we adopt a standard denoiser-based diffusion parameterization of the conditional score (Karras et al., 2022) and denote the corresponding denoiser by $D_\theta(\boldsymbol{z}_t, t, \boldsymbol{c})$. For small $\sigma_c$, linearizing $D_\theta$ with respect to $\boldsymbol{c}_{\text{low}}$ via a first-order Taylor expansion yields an Euler discretization of the probability flow ODE of the form:

$$\begin{aligned}\boldsymbol{z}_{t-1} &\approx \boldsymbol{z}_t + \rho_t \nabla_{\boldsymbol{z}_t} \log p_t(\boldsymbol{z}_t \mid \boldsymbol{c}) + \boldsymbol{\eta}_t^\top \boldsymbol{n}, \\ \boldsymbol{\eta}_t &:= \frac{\rho_t \sigma_c}{\sigma(t)^2} \nabla_{\boldsymbol{c}_{\text{low}}} D_\theta(\boldsymbol{z}_t, t, \boldsymbol{c}).\end{aligned} \quad (4)$$

This mirrors the Langevin-like decomposition of CADS,

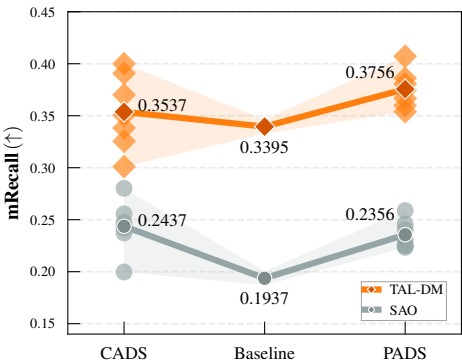

Figure 4. Overall average of genre-wise mRecall on MelBench; the mean $\mathrm{IPR}_{CLAP\text{-}f}$ Recall. Markers: different parameter settings; Lines: method-level averages.

but with stochasticity induced solely by the low-salience subspace (see Appendix B.3). By confining CADS-style conditioning noise to $c_{\mathrm{low}}$, PADS induces Langevin-style exploration while avoiding first-order variation along salient semantic coordinates, thereby preserving prompt semantics. In other words, Eq. (4) describes a subspace-restricted form of CADS; depending on how broadly one defines the low-salience subspace, CADS is recovered as the special case where the selected subspace spans the full conditioning.

### 3.2. Text-Aware Latent Space

In real-world data curation, a single audio can support multiple plausible interpretations, and datasets typically label it with one that matches the collection goal. Accordingly, training data should be understood not as a mere set of audio samples, but as a joint text–audio distribution that encodes an intended semantic structure. However, in the standard LDM pipeline, the VAE learns the latent space using audio only, and the DM performs conditional generation in a text-unaware latent space. In this setting, there is little guarantee that the neighborhood structure is aligned with the dataset's intended text semantic structure, particularly for global attributes such as genre. As a result, even with conditioning perturbation, it is generally not possible to stably obtain diverse outputs while preserving the intended genre, making reliable genre-wise coverage difficult to achieve.

Fig. 4 reports genre-wise coverage, measuring how broadly samples generated from genre-specific prompts cover each genre's reference manifold, across various perturbation settings. Conditioning perturbation tends to expand text-aligned diversity and may, as a byproduct, increase genre-consistent diversity, but the absolute level remains low, indicating that diversity satisfying the genre condition is still insufficient. To mitigate this, we redesign the VAE stage of the LDM to be text-aware, encouraging the resulting latent representations to better reflect text semantic structure. We define TAL as a selected subset of these representations, where the DM is trained for sampling.

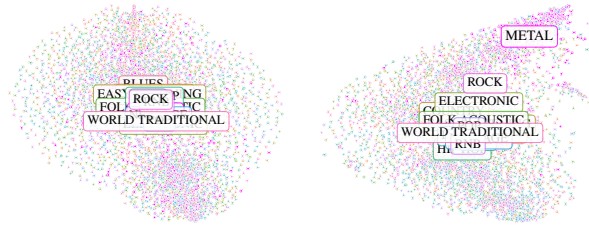

(a) Audio Latent space     (b) Text-Aware Latent space

Figure 5. t-SNE visualization of genre embeddings on the Mel-Bench dataset, with genre centroid labels. We also report kNN precision@k ($\uparrow$); the chance level is 0.067: (a) Prec@5/10/20 = 0.066/0.068/0.067, (b) Prec@5/10/20 = 0.243/0.211/0.189.

To this end, we are inspired by prior MoE-style multimodal VAE designs (Shi et al., 2019; Sutter et al., 2020; Palumbo et al., 2023) and adopt a shared–private architecture that preserves modality-specific information for audio while encouraging semantic factors to be consistent across modalities. Concretely, we employ private encoders $q_{\phi_{\boldsymbol{w}}}$ and shared encoders $q_{\phi_{\boldsymbol{z}}}$, construct the latents by concatenating their outputs in various combinations, and perform reconstruction using an audio decoder $p_{\theta_m}$ and a text decoder $p_{\theta_p}$. We denote the private latents for audio and text as $\boldsymbol{w}_m$ and $\boldsymbol{w}_p$, and the shared latents as $\boldsymbol{z}_m$ and $\boldsymbol{z}_p$, respectively (details in Appendix C.2).

In SAO (Evans et al., 2025), the VAE is trained using audio only, with the objective defined as a weighted sum of multiple loss terms:

$$\mathcal{L}_{VAE} = \alpha_{adv}\mathcal{L}_{adv} + \alpha_{mrstft}\mathcal{L}_{mrstft} + \alpha_{kl}\mathcal{L}_{kl}. \quad (5)$$

Here, $\mathcal{L}_{adv}$ is an adversarial loss that includes feature matching and uses the discriminator from Encodec (Défossez et al., 2023). The term $\mathcal{L}_{mrstft}$ is a reconstruction loss based on perceptually weighted multi-resolution STFT (Steinmetz & Reiss, 2020), and $\mathcal{L}_{kl}$ is a KL divergence term that regularizes the VAE.

We retain $\mathcal{L}_{mrstft}$ for audio reconstruction, and introduce an additional cross-entropy loss for text reconstruction, computed between the ground-truth text sample $\boldsymbol{x}_p$ and the generated sample $\tilde{\boldsymbol{x}}_p$, denoted as $\mathcal{L}_{ce}(\boldsymbol{x}_p, \tilde{\boldsymbol{x}}_p)$. Moreover, we propose a Latent Alignment (LA) loss that encourages the audio-side and text-side shared-latent candidates to encode consistent semantics by pulling their mean vectors closer:

$$\mathcal{L}_{LA}(\boldsymbol{z}_m, \boldsymbol{z}_p) = \frac{1}{D}\|\mu_{\boldsymbol{z}_m} - \mu_{\boldsymbol{z}_p}\|_2^2. \quad (6)$$

Here, $D$ denotes the dimensionality of the shared-latent mean vectors. Inspired by the MoE-style posterior aggregation in a factorized shared–private latent space (Eq. (36) in Appendix C.1), we train our model using a practical weighted loss that combines audio/text reconstruction, KL

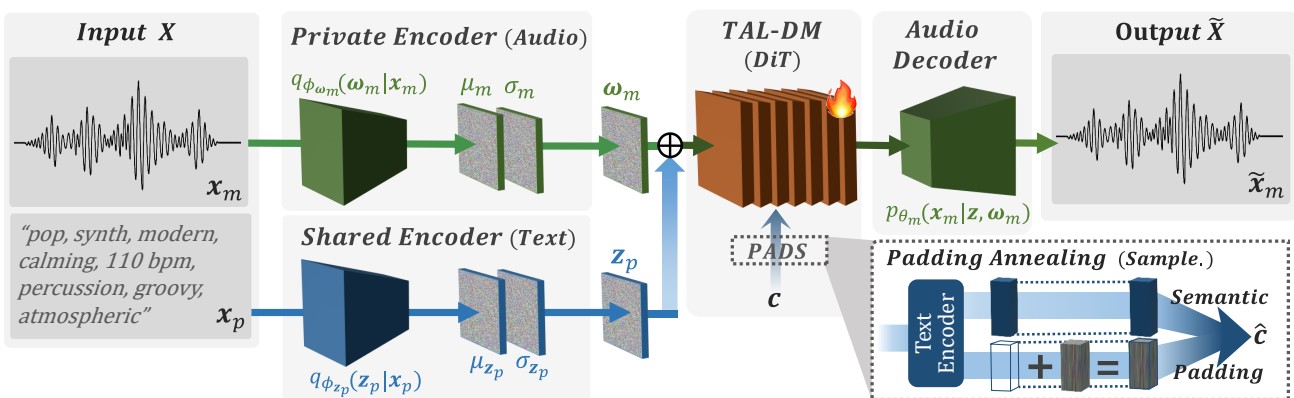

*Figure 6.* The training pipeline of the diffusion model in TAL, together with PADS applied only at sampling time. The DM is trained to predict the noise residual in the TAL latent space. ($\oplus$ : concatenation)

regularization, and the proposed $\mathcal{L}_{LA}$:

$$
\begin{aligned}
\mathcal{L}_{MoE\text{-}mVAE} = \ & \alpha_{adv}\mathcal{L}_{adv} + \alpha_{LA}\mathcal{L}_{LA} \\
& + \underbrace{\alpha_{mrstft}\left[\tfrac{1}{2}\mathcal{L}_{mrstft}^m + \tfrac{1}{2}\mathcal{L}_{mrstft}^p\right]}_{\text{Audio Reconstruction}} \\
& + \underbrace{\alpha_{ce}\left[\tfrac{1}{2}\mathcal{L}_{ce}^m + \tfrac{1}{2}\mathcal{L}_{ce}^p\right]}_{\text{Text Reconstruction}} \qquad (7)\\
& + \underbrace{\sum_{u\in\{z_m,z_p,w_m,w_p\}}\mathcal{L}_{kl}(q_{\phi_u}).}_{\text{Regularization}}
\end{aligned}
$$

The audio and text reconstruction terms are specified by the encoders–decoder combinations as follows, where the shared and private latent outputs from the selected encoders are concatenated and passed to the decoder:

$$
\begin{aligned}
\mathcal{L}_{mrstft}^m &:= \mathcal{L}_{mrstft}(q_{\phi_{z_m}}, q_{\phi_{w_m}}, p_{\theta_m}), \\
\mathcal{L}_{mrstft}^p &:= \mathcal{L}_{mrstft}(q_{\phi_{z_p}}, q_{\phi_{w_m}}, p_{\theta_m}), \\
\mathcal{L}_{ce}^m &:= \mathcal{L}_{ce}(q_{\phi_{z_m}}, q_{\phi_{w_p}}, p_{\theta_p}), \\
\mathcal{L}_{ce}^p &:= \mathcal{L}_{ce}(q_{\phi_{z_p}}, q_{\phi_{w_p}}, p_{\theta_p}).
\end{aligned}
\qquad (8)
$$

We define TAL as the concatenated latent representation formed by the private audio latent $w_m$ and the shared text latent $z_p$. To provide an intuitive view of how TAL captures genre information, we construct TAL latents from paired audio–text samples and visualize them by genre (Fig. 5). In TAL, samples within the same genre tend to be more spatially clustered, and different genres exhibit a partially separable geometric structure. Beyond this qualitative visualization, we quantify neighborhood-level genre coherence using kNN precision@k, defined as the fraction of the $k$ nearest neighbors (by $\ell_1$-distance) in the latent space that share the same genre label. As shown in Fig. 5, TAL substantially improves Prec@5/10/20 over the audio-only latent space, indicating that TAL's local neighborhoods are more genre-faithful and thus supporting its objective of aligning

semantically similar audio–text pairs to be nearby in latent space (details in Appendix D).

### 3.3. A Unified Pipeline with TAL and PADS

In the TAL representation space, the DM learns the denoising process, and PADS is applied at sampling time. The overall pipeline is summarized in Fig. 6.

For TAL-based DM training, we use only the private audio encoder, which captures detailed acoustic information, and the shared text encoder, which produces the shared latent from text conditions. Because MoE-mVAE training applies the LA loss Eq. (6) to encourage the posteriors of the shared latents across audio and text to be similar, the shared latent can be stably constructed from text encoding alone. Accordingly, the DM is trained to denoise in the TAL space formed by concatenating the private audio latent with the text-derived shared latent, encouraging the resulting denoised latents to remain consistent with text semantics, particularly global attributes such as genre. At sampling time, the trained TAL-DM promotes sampling within text-conditioned latent neighborhoods as illustrated in Fig. 4, thereby structurally mitigating genre mismatch.

We further combine PADS during sampling to stably induce text-aligned diversity by expanding exploration while preserving prompt semantics. From the genre perspective, Fig. 4 shows that CADS can broaden the sample distribution via perturbation and may occasionally include samples near parts of the reference manifold that were previously uncovered, increasing genre coverage. However, such improvements are erratic across perturbation parameter settings, leading to inconsistent gains. In contrast, PADS provides additional exploration freedom while suppressing direct corruption of semantic embeddings, thereby acquiring diverse samples in the local neighborhood while maintaining text alignment. As a result, PADS tends to improve recall more consistently under genre conditions. In summary, within the proposed unified pipeline, TAL aligns the space in which

sampling occurs with text semantic structure to strengthen global consistency, and PADS then structurally constrains where perturbations are applied to expand diversity in a stable manner.

# 4. Experiments

Our proposed pipeline adopts the SAO (Evans et al., 2025) diffusion model architecture, and our sampling implementation, including PADS, is built on Stable Audio Tools (Stability AI, 2024d). We design our training data configuration to match the public SAO setup as closely as possible, and add a small in-house music set (excluded from evaluation) to increase the proportion of music during training and improve music-focused reproducibility under T2M perturbation. For a fair comparison, we train the SAO baseline under the same training data configuration, and both our model and the SAO baseline are trained from scratch without using pretrained weights. For evaluation, to reduce confounding factors and keep the evaluation setting consistent, we focus on instrumental generation (no vocals) with clips at least 30 seconds long. Therefore, we use SongDescriber (Manco et al., 2023) as-is, and prepare an instrumental subset of MelBench (Chowdhury et al., 2024) by removing vocal tracks and sampling uniformly across genres. We further standardize generation conditions by using tag-based prompts throughout (see Appendix E).

For training, MoE-mVAE is trained with batch size 4 on an 8×H100 setup for ∼250 hours, and the DiT is trained in the same environment with batch size 4 for ∼320 hours. For generation, all samples are generated with 100 sampling steps, $\omega_{cfg}$=7.0, and DPM-Solver++ (Lu et al., 2025). Before presenting the main results, Sec. 4.1 summarizes the evaluation metrics used in this paper, including our proposed alignment metric. We then present the main results for the proposed methods in Secs. 4.2 to 4.5.

## 4.1. Metrics

**Fidelity.** Generation quality is measured following the Stable Audio Metrics (Stability AI, 2024c) setup, using SongDescriber embedding-based $KL_{passt}$ and $FD_{openl3}$.

**Text Alignment.** Text–audio alignment is evaluated using CLAP (Wu et al., 2023), but to reduce score variance caused by segment selection and to reflect musical form, we propose and use a CLAP-fixed (CLAP-f) protocol that computes embeddings at multiple fixed positions within a sample. It preserves the overall behavior of standard CLAP while reducing evaluation randomness, enabling more stable measurement (see Appendix F.1).

**Diversity and Coverage.** We evaluate diversity using the Vendi score (Friedman & Dieng, 2023), IPR Recall (Kynkäänniemi et al., 2019), and the In-Batch Similarity

*Table 1.* Comparison of CADS and PADS for instrumental music generation on SongDescriber across T2M frameworks. (**Pert.**: perturbation strategy)

| Model | Pert. | Vendi ↑ | CLAP-f↑ | $KL_{passt}$↓ |
|---|---|---|---|---|
| ACE-Step | - | 16.6154 | 0.2425 | 0.7456 |
| | **CADS** | **23.4070** | 0.1981 | 0.9240 |
| | | △ +6.7916 | ▽ −0.0444 | △ +0.1784 |
| | **PADS** | **23.3399** | 0.2318 | 0.7918 |
| | | △ +6.7245 | ▽ −0.0107 | △ +0.0462 |
| MelodyFlow | - | 18.2190 | 0.2770 | 0.6873 |
| | **PADS** | **21.5843** | 0.2557 | 0.7671 |
| | | △ +3.3653 | ▽ −0.0213 | △ +0.0798 |

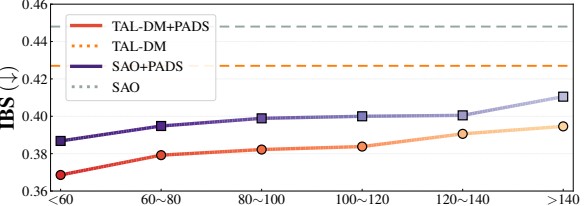

*Figure 7.* Effect of prompt length range on in-batch similarity for SongDescriber.

(IBS) score (Corso et al., 2024), all computed in the CLAP-f embedding space. Vendi and IPR are computed in a prompt-pooled manner over all generated samples: Vendi quantifies intrinsic dispersion of the generated distribution without a reference set, whereas Recall measures how broadly generated samples cover a reference distribution. For $IPR_{CLAP\text{-}f}$ (IPR Recall computed using CLAP-f embeddings), extracting multiple segment embeddings from the same sample can make the $k$-NN–based radius overly small, leading to unstable estimates; we therefore compute IPR Recall per segment and report the final score as the average across segments (Appendix F.2). Complementary to these prompt-pooled metrics, IBS measures within-prompt diversity by computing similarity among samples generated from the fixed prompt.

## 4.2. Prompt Length Controls the Diversity Budget

Fig. 7 shows that PADS improves diversity even within a fixed prompt, as reflected by reduced in-batch similarity. Moreover, because PADS perturbs only padding positions, shorter prompts leave a larger padding subspace for annealed exploration, which can lead to slightly higher within-prompt diversity. This prompt-length effect aligns with typical user intent: short prompts invite varied interpretations, while longer prompts emphasize tighter adherence. Even for very long prompts with little padding, we enforce a minimum perturbation budget so that PADS injects a non-trivial amount of exploration noise.

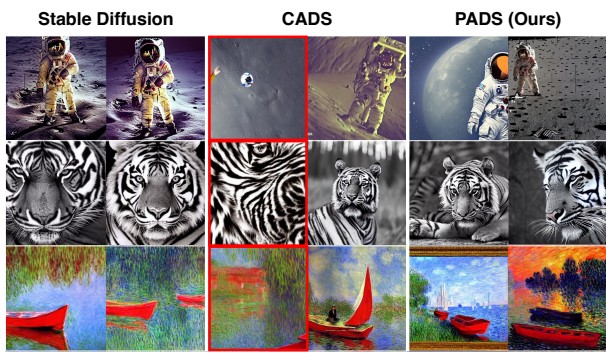

*Figure 8.* Images generated by Stable Diffusion v1.5 using identical parameters across all experiments. Prompts: "An astronaut on the Moon", "A grayscale tiger", "Monet-style painting of a red boat".

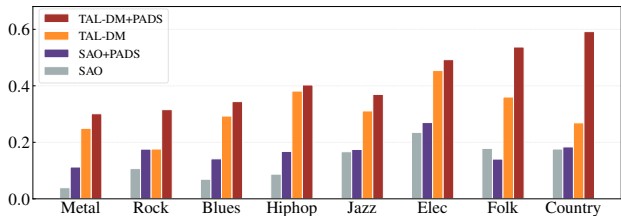

*Figure 9.* Genre-wise $\text{IPR}_{CLAP\text{-}f}$ Recall on MelBench.

### 4.3. Generalization of PADS

To test the generalization of PADS, we apply it to other diffusion-based T2M models that support instrumental generation and produce 30-second-or-longer audio clips. In Table 1, ACE-Step (Gong et al., 2025)[1] preserves text alignment at comparable Vendi levels relative to prior methods. For MelodyFlow (Le Lan et al., 2024), CADS does not increase diversity, likely because it hurts fidelity: outputs become noise-like, so the Vendi score no longer improves, whereas PADS yields stable diversity gains.

Fig. 8 further reports results on a T2I model that retains padding embeddings as part of the conditioning sequence (i.e., not fully masked out). CADS occasionally produces misaligned text–image samples (marked in red), while PADS maintains text alignment while increasing diversity.

### 4.4. Genre-wise Coverage Gains from TAL

Fig. 9 compares per-genre coverage for DMs trained in TAL. TAL-based models achieve higher Recall than SAO for most genres, with especially large gains for genres with distinctive acoustic and structural traits, such as Metal.

### 4.5. Ablation of TAL and PADS

Table 2 reports an ablation of our unified pipeline, comparing the diffusion model's training latent space and the conditioning perturbation strategy while matching CLAP-f

---

[1]For instrumental generation, we add "[inst]" to the lyric conditioning field, following the baseline setup.

*Table 2.* Ablation of TAL and PADS on MelBench and SongDescriber under a matched CLAP-f level ($\approx 0.32$). (**Pert.**: perturbation strategy, mRecall: mean $\text{IPR}_{CLAP\text{-}f}$ Recall)

| Architecture | | Evaluation Dataset | | | |
|---|---|---|---|---|---|
| Latent Space | Pert. | MelBench | SongDescriber | | |
| | | mRecall↑ | Vendi ↑ | $\text{KL}_{passt}$ ↓ | $\text{FD}_{openl3}$ ↓ |
| Audio | **CADS** | 0.237 | 22.591 | 0.669 | 147.730 |
| Audio | **PADS** | 0.212 | 24.367 | 0.652 | 139.838 |
| TAL | **CADS** | 0.391 | 23.873 | 0.669 | 145.393 |
| TAL | **PADS** | **0.407** | **26.075** | **0.626** | **135.542** |

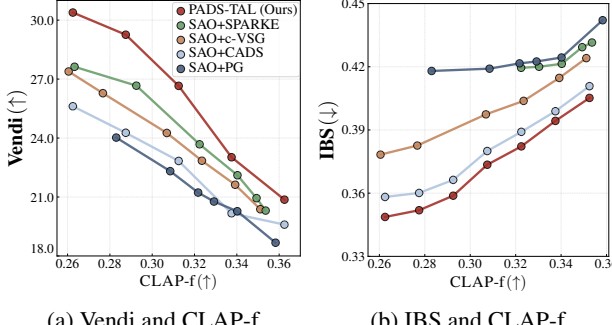

(a) Vendi and CLAP-f    (b) IBS and CLAP-f

*Figure 10.* Diversity versus text alignment measured by CLAP-f across methods: (a) Vendi, (b) IBS. Each curve is obtained by varying the hyperparameters of the corresponding method.

to a similar level ($\approx 0.32$). Both CADS and PADS are designed to increase diversity, but their alignment–diversity trade-offs differ; accordingly, we compare strategies at a matched CLAP-f level. Under this controlled alignment level, our pipeline achieves the strongest diversity, improving overall diversity by 15.4% and within-genre diversity by 71.6% relative to applying CADS to a diffusion model trained in the audio-only latent space; at the same time, it also yields the best fidelity when assessed by multiple fidelity metrics.

### 4.6. Comparison with Diversity Enhancement Methods

To the best of our knowledge, CADS is the only prior method that explicitly increases diversity by injecting perturbations into the conditioning signal at diffusion sampling time. Nevertheless, to provide a broader empirical context, we additionally present Fig. 10, which compares PADS-TAL with several related sampling-time diversity enhancement methods. Since these methods were originally studied on T2I models, we adapt them to SAO and evaluate them in the T2M setting. Specifically, we sweep the hyperparameters of each method to trace the text-alignment–diversity trade-off curve (details in Appendix G.7), evaluating diversity in terms of both within-prompt IBS and sample-level Vendi.

Overall, we observe that the relative ordering of baselines

*Table 3.* Human evaluation results (MOS; 1–5) collected from 136 participants in a blind setting: (a) within-model comparison and (b) across-model comparison.

| | | Quality | Diversity | Alignment |
|---|---|---|---|---|
| (a) | SAO+CADS | 3.18 | 3.47 | 3.09 |
| | SAO+PADS | **3.89** | **3.49** | **3.79** |

| | | Quality | Diversity | Alignment |
|---|---|---|---|---|
| (b) | SAO | **3.81** | 3.41 | **3.83** |
| | SAO+CADS | 3.36 | 3.56 | 2.87 |
| | PADS-TAL | 3.79 | **3.71** | 3.78 |

reported in prior work does not always transfer when these methods are applied to latent diffusion models for audio generation. In the practically usable regime, PADS-TAL achieves improved diversity while maintaining stronger prompt alignment.

### 4.7. Human Evaluation

Table 3 summarizes the human evaluation results in terms of Mean Opinion Score (MOS), collected in a blind setting through a manually created survey page with 136 participants. Participants rated quality, diversity, and text alignment on a 1–5 scale. To assess the qualitative effect of each sampling strategy in a controlled manner, the within-model comparison was conducted on SAO with the prompt and initial noise fixed across methods. In contrast, the across-model comparison used different prompts and initial random seeds for different models. Overall, the results indicate that PADS and TAL remain effective under human evaluation, maintaining high generation quality while improving diversity.

## 5. Limitation

Prior work (Palumbo et al., 2023) highlights that, when modality-specific latent subspaces exist, the shared subspace can be under-utilized and reconstructions may rely disproportionately on private pathways. Our MoE-mVAE is similarly designed to leverage shared information while maintaining audio reconstruction performance; however, we observe that the learned latents may not always be fully reflected in the generated outputs. As a direction for future work, one could strengthen the coupling between the learned latents and the decoders—for example, by adding regularization or auxiliary objectives that encourage the decoders to utilize the learned latents more consistently.

## 6. Conclusion

In this work, we emphasize that diversity in T2M diffusion models is not merely about increasing sample dispersion, but about generating varied outputs within the range that

preserves prompt semantics, namely text-aligned diversity. We further show that indiscriminately perturbing the entire condition does not resolve genre mismatch, and generations can still deviate from prompt-specified genre attributes. To address this, we propose two complementary components, PADS and TAL. PADS is an inference-time strategy that confines conditioning perturbations to a low-salience subspace, expanding exploration while minimizing corruption of semantic cues. TAL aligns local latent neighborhoods with global text semantics such as genre, providing a more reliable representation for genre-consistent generation. Our unified pipeline improves diversity both across prompts and within a single prompt while maintaining text alignment and producing more genre-consistent generations.

## Acknowledgements

We thank the reviewers and area chair for their constructive feedback. We also thank our colleagues for helpful discussions. This work was supported by the National Research Foundation of Korea (NRF) grant funded by the Korean government (MSIT) (No. RS-2024-00456152) and the "Advanced GPU Utilization Support Program" funded by the Government of the Republic of Korea (Ministry of Science and ICT), and the authors gratefully acknowledge the Cluster Server for Computational Science at Pusan National University for providing computational resources.

## Impact Statement

Our work improves text-aligned diversity while preserving genre consistency for diffusion-based T2M generation, which can benefit creative workflows by enabling broader exploration while maintaining semantic alignment and audio quality. We emphasize responsible data and evaluation practices by training primarily on openly licensed audio (CC0, CC BY, and CC Sampling+) and an in-house dataset (approximately 20k tracks) curated under appropriate usage rights, and by evaluating only on public benchmarks. As with other music generation methods, potential downstream risks include deceptive use and outputs that may resemble training examples depending on deployment and prompting. We recommend responsible use, including disclosure of synthetic audio, safeguards against impersonation/deception, and continued attention to licensing and provenance in any future release.

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

# Appendix

## Contents

## A. Sampling Configuration for CADS and PADS

Since CADS and PADS share similar hyperparameters, we summarize the sampling configurations used for both perturbation schemes. Detailed configurations and hyperparameters for all experiments reported in the main text (including the referenced figures and tables) are provided in Table 17.

### A.1. CADS and PADS Parameters

Both CADS and PADS share the cutoff thresholds $\tau_1$ and $\tau_2$ that define the annealing interval, as well as the scheduler type used for the annealing schedule. In addition, both methods use a noise scale $s$ that controls the magnitude of injected noise. CADS further includes a mixing factor $\psi$ for forming a weighted sum with the rescaled condition.

Concretely, for the annealing interval we evaluated combinations of $\tau_1 \in \{0.1, 0.4, 0.6, 0.8, 0.9, 0.97\}$ and $\tau_2 \in \{0.85, 0.9, 0.97, 1.0\}$. For CADS, we used the piecewise linear scheduler reported as the best-performing choice in the original work, faithfully reproducing its setting. For PADS, we considered three scheduler families—polynomial, cosine, and step—and, after evaluation, standardized on a first-order polynomial (i.e., linear) scheduler for the main experiments. Additional details on the scheduler types are provided in Appendix A.2.

For the noise scale applied to the padding region, we searched within the range $s \in \{0.01, 0.06, 0.1, 0.25, 0.35, 0.5\}$, focusing on values commonly used in prior work. We include CADS with the rescaling strategy proposed in the original paper, following its configuration as closely as possible, whereas PADS does not apply rescaling. For CADS, the mixing factor $\psi$ used to combine the rescaled condition with the original condition was fixed to 1.0, following the recommendation of the original paper and aligning with our goal of stable sampling with more diverse generation.

### A.2. Annealing Scheduler for PADS

While CADS uses the piecewise linear scheduler that performed best in the original work, we analyze how the choice of annealing scheduler affects PADS and select a scheduler suitable for text-aligned diversity without degrading text alignment. Following Fig. 11, we comprehensively compared polynomial, cosine, and step annealing schedules by varying the exponent $d$ for the polynomial and cosine schedulers. Each scheduler is defined as follows:

$$\gamma_{\text{polynomial}}(t) = \begin{cases} 1, & \text{if } 0 \leq t \leq \tau_1, \\ \left(\dfrac{\tau_2 - t}{\tau_2 - \tau_1}\right)^d, & \text{if } \tau_1 < t < \tau_2, \\ 0, & \text{if } \tau_2 \leq t \leq 1. \end{cases} \tag{9}$$

$$\gamma_{\text{cosine}}(t) = \begin{cases} 1, & \text{if } 0 \leq t \leq \tau_1, \\ \left[\cos\left(\dfrac{\pi}{2}\dfrac{t - \tau_1}{\tau_2 - \tau_1}\right)\right]^d, & \text{if } \tau_1 < t < \tau_2, \\ 0, & \text{if } \tau_2 \leq t \leq 1. \end{cases} \tag{10}$$

$$\gamma_{\text{step}}(t) = \begin{cases} 1, & \text{if } 0 \leq t \leq \tau_1, \\ 0, & \text{if } \tau_1 < t \leq 1. \end{cases} \tag{11}$$

As shown in Table 4, the CLAP-f score varies with the shape of the annealing curve; in particular, schedulers that retain conditional information for a longer portion of the sampling trajectory tend to yield higher CLAP-f scores. Overall, we find that choosing the linear schedule (i.e., polynomial with $d = 1$) maintains stronger conditional consistency (as measured by CLAP-f) while achieving competitive diversity compared to other scheduling strategies. Based on these results—and to ensure a principled comparison with CADS—we primarily adopt the linear scheduler in our experiments.

## B. Padding-Annealed Diffusion Sampling Details

In this section, motivated by EDM (Karras et al., 2022) and the Langevin-like view of condition perturbations in CADS, we present the intuition behind padding annealing and summarize practical considerations for text-conditioning formats, including zero-padded embeddings and masked attention.

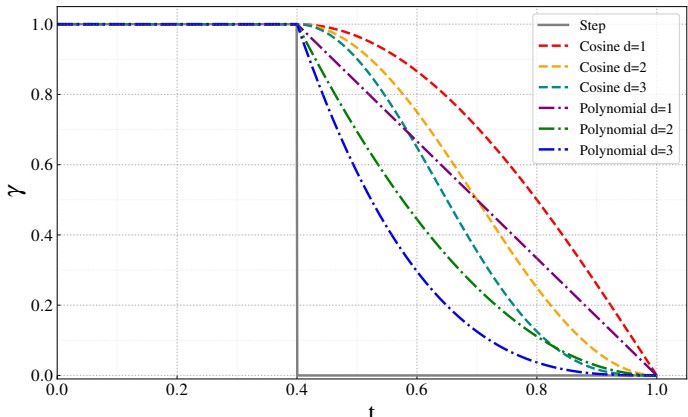

*Figure 11.* Annealing curves for schedulers.

*Table 4.* Comparison of annealing schedulers for SAO with PADS on SongDescriber. ($\tau_1 = 0.6$, $\tau_2 = 0.9$, $s = 0.25$).

| Scheduler | CLAP-f | Vendi |
|---|---|---|
| Step | 0.312 | 24.843 |
| Poly ($d = 3$) | 0.330 | 23.487 |
| Cos ($d = 2$) | 0.332 | 23.351 |
| Cos ($d = 3$) | 0.334 | 24.384 |
| Poly ($d = 2$) | 0.334 | 23.038 |
| Cos ($d = 1$) | 0.334 | 23.234 |
| **Poly ($d = 1$)** | **0.335** | 23.230 |

## B.1. Background on EDM Sampling and Denoiser Parameterization

EDM (Karras et al., 2022) introduces a noise schedule $\sigma(t)$ with $\sigma(0) = 0$ and $\sigma(1) = \sigma_{\max}$, together with an optional signal scaling $s(t)$, and formulates the probability flow ODE in terms of the noise-level-dependent marginals $p(z; \sigma)$ as

$$\mathrm{d}\boldsymbol{z}_t = \left[ \frac{\dot{s}(t)}{s(t)} \boldsymbol{z}_t - s(t)^2 \dot{\sigma}(t)\sigma(t) \nabla_{\boldsymbol{z}_t} \log p\left( \frac{\boldsymbol{z}_t}{s(t)}; \sigma(t) \right) \right] \mathrm{d}t, \tag{12}$$

where the dot denotes a time derivative and $p(\boldsymbol{z}; \sigma)$ denotes the marginal distribution obtained by Gaussian corruption with standard deviation $\sigma$. In particular, $p(\boldsymbol{z}; 0) = p_{\mathrm{data}}$, and $p(\boldsymbol{z}; \sigma_{\max})$ corresponds to $p_{\mathrm{data}}$ convolved with $\mathcal{N}(\boldsymbol{0}, \sigma_{\max}^2 \mathbf{I})$. For generation, we initialize at high noise $\boldsymbol{z}_{t=1} \sim \mathcal{N}(\boldsymbol{0}, \sigma_{\max}^2 \mathbf{I})$ (when $\sigma_{\max} \gg \sigma_{\mathrm{data}}$) and integrate the ODE backward in time from $t = 1$ to $t = 0$.

For the representative setting recommended in EDM, $s(t) = 1$, the probability flow ODE simplifies to

$$\mathrm{d}\boldsymbol{z}_t = -\dot{\sigma}(t)\sigma(t) \nabla_{\boldsymbol{z}_t} \log p(\boldsymbol{z}_t; \sigma(t)) \, \mathrm{d}t. \tag{13}$$

To numerically solve Eq. (13) along a discrete noise grid $\{\sigma_t\}_{t=0}^T$ with $\sigma_0 = 0$ and $\sigma_T = \sigma_{\max}$, we integrate the ODE backward (from $t = T$ to $t = 0$) using Euler discretization:

$$\boldsymbol{z}_{t-1} = \boldsymbol{z}_t + \rho_t \nabla_{\boldsymbol{z}_t} \log p(\boldsymbol{z}_t; \sigma_t), \qquad \rho_t := \sigma_t(\sigma_t - \sigma_{t-1}), \tag{14}$$

where $\rho_t > 0$ since $\sigma_{t-1} < \sigma_t$ along the (backward) denoising trajectory.

EDM further parameterizes the network to predict a denoised estimate rather than the score directly. Let $D_\theta(\boldsymbol{z}, \sigma) \approx \mathbb{E}[\boldsymbol{x}_0 \mid \boldsymbol{z}]$ denote the denoiser under Gaussian corruption $\boldsymbol{z} = \boldsymbol{x}_0 + \sigma\boldsymbol{\epsilon}$ with $\boldsymbol{\epsilon} \sim \mathcal{N}(\boldsymbol{0}, \mathbf{I})$. Then, using the denoising-score identity employed in EDM (Karras et al., 2022), the score of the noisy marginal can be written as

$$s_\theta(\boldsymbol{z}, \sigma) \approx \nabla_{\boldsymbol{z}} \log p(\boldsymbol{z}; \sigma) = \frac{D_\theta(\boldsymbol{z}, \sigma) - \boldsymbol{z}}{\sigma^2}. \tag{15}$$

In practice, we compute $s_\theta(\boldsymbol{z}, \sigma)$ via Eq. (15) and substitute it into the discretized update in Eq. (14) (or its stochastic counterpart, e.g., Euler–Maruyama) to advance the sampling trajectory.

## B.2. Background on the Langevin-like Interpretation of CADS

CADS perturbs the conditioning signal by adding Gaussian noise. When the injected noise is small relative to the conditioning magnitude, the corrupted condition can be written as

$$\hat{\boldsymbol{c}} = \boldsymbol{c} + \Delta \boldsymbol{c}, \qquad \Delta \boldsymbol{c} := \sigma_c \boldsymbol{n}, \qquad \boldsymbol{n} \sim \mathcal{N}(\boldsymbol{0}, \mathbf{I}), \tag{16}$$

where $\sigma_c$ denotes the condition-noise scale.

Following the EDM denoiser parameterization (Eq. (15)), we express the conditional score using the corresponding conditional noisy marginal,[2]

$$\nabla_{\boldsymbol{z}_t} \log p_t(\boldsymbol{z}_t \mid \boldsymbol{c}) \approx \frac{D_\theta(\boldsymbol{z}_t, t, \boldsymbol{c}) - \boldsymbol{z}_t}{\sigma(t)^2}. \tag{17}$$

For a small perturbation $\Delta\boldsymbol{c}$, a first-order Taylor expansion of the denoiser with respect to the conditioning yields

$$D_\theta(\boldsymbol{z}_t, t, \hat{\boldsymbol{c}}) \approx D_\theta(\boldsymbol{z}_t, t, \boldsymbol{c}) + \nabla_{\boldsymbol{c}} D_\theta(\boldsymbol{z}_t, t, \boldsymbol{c})^\top \Delta\boldsymbol{c}, \tag{18}$$

where $\nabla_{\boldsymbol{c}} D_\theta(\boldsymbol{z}_t, t, \boldsymbol{c})$ denotes the Jacobian of the denoiser output with respect to $\boldsymbol{c}$ (flattening $L \times D$ into a vector when needed).

To analyze the effect of condition perturbation on the sampling dynamics, we substitute the perturbed condition $\hat{\boldsymbol{c}}$ into the Euler update of the probability flow ODE in Eq. (14), i.e., $\boldsymbol{z}_{t-1} = \boldsymbol{z}_t + \rho_t \nabla_{\boldsymbol{z}_t} \log p_t(\boldsymbol{z}_t \mid \hat{\boldsymbol{c}})$. Using the EDM-style denoiser-to-score relation for the conditional score in Eq. (17), together with the first-order expansion in Eq. (18), we obtain the Langevin-like decomposition:

$$\begin{aligned}
\boldsymbol{z}_{t-1} &\approx \boldsymbol{z}_t + \rho_t \frac{D_\theta(\boldsymbol{z}_t, t, \hat{\boldsymbol{c}}) - \boldsymbol{z}_t}{\sigma(t)^2} \\
&\approx \boldsymbol{z}_t + \rho_t \frac{D_\theta(\boldsymbol{z}_t, t, \boldsymbol{c}) - \boldsymbol{z}_t}{\sigma(t)^2} + \rho_t \frac{1}{\sigma(t)^2} \nabla_{\boldsymbol{c}} D_\theta(\boldsymbol{z}_t, t, \boldsymbol{c})^\top \Delta\boldsymbol{c} \\
&= \boldsymbol{z}_t + \rho_t \nabla_{\boldsymbol{z}_t} \log p_t(\boldsymbol{z}_t \mid \boldsymbol{c}) + \frac{\rho_t \sigma_c}{\sigma(t)^2} \nabla_{\boldsymbol{c}} D_\theta(\boldsymbol{z}_t, t, \boldsymbol{c})^\top \boldsymbol{n}.
\end{aligned} \tag{19}$$

Defining

$$\boldsymbol{\eta}_t := \frac{\rho_t \sigma_c}{\sigma(t)^2} \nabla_{\boldsymbol{c}} D_\theta(\boldsymbol{z}_t, t, \boldsymbol{c}), \tag{20}$$

Eq. (19) can be written compactly as

$$\boldsymbol{z}_{t-1} \approx \boldsymbol{z}_t + \rho_t \nabla_{\boldsymbol{z}_t} \log p_t(\boldsymbol{z}_t \mid \boldsymbol{c}) + \boldsymbol{\eta}_t^\top \boldsymbol{n}. \tag{21}$$

In particular, the update consists of (i) a *drift term* $\nabla_{\boldsymbol{z}_t} \log p_t(\boldsymbol{z}_t \mid \boldsymbol{c})$ that follows the conditional score field under the unperturbed condition $\boldsymbol{c}$, and (ii) an additional *stochastic term* $\boldsymbol{\eta}_t^\top \boldsymbol{n}$ induced by the conditioning noise, whose covariance is shaped by the denoiser's local sensitivity $\nabla_{\boldsymbol{c}} D_\theta(\boldsymbol{z}_t, t, \boldsymbol{c})$.

### B.3. Intuition behind PADS from a Langevin-like Perspective

Motivated by the Langevin-like interpretation of condition perturbations in CADS, we design PADS to inject exploration noise only through a low-salience subspace of the conditioning. Specifically, we decompose the conditioning sequence into a salient part and a low-salience part:

$$\boldsymbol{c} = \begin{bmatrix} \boldsymbol{c}_{\text{sal}}; \ \boldsymbol{c}_{\text{low}} \end{bmatrix}, \tag{22}$$

and keep $\boldsymbol{c}_{\text{sal}}$ fixed while perturbing only $\boldsymbol{c}_{\text{low}}$:

$$\hat{\boldsymbol{c}} = \begin{bmatrix} \boldsymbol{c}_{\text{sal}}; \ \hat{\boldsymbol{c}}_{\text{low}} \end{bmatrix}, \qquad \hat{\boldsymbol{c}}_{\text{low}} = \boldsymbol{c}_{\text{low}} + \Delta\boldsymbol{c}_{\text{low}}, \qquad \Delta\boldsymbol{c}_{\text{low}} := \sigma_c \boldsymbol{n}, \quad \boldsymbol{n} \sim \mathcal{N}(\boldsymbol{0}, \mathbf{I}), \tag{23}$$

where $\sigma_c$ controls the noise level injected into the low-salience component and $\boldsymbol{n}$ has the same shape as $\boldsymbol{c}_{\text{low}}$.

When $\Delta\boldsymbol{c}_{\text{low}}$ is small, a first-order Taylor expansion of the denoiser with respect to the low-salience component gives

$$\begin{aligned}
D_\theta(\boldsymbol{z}_t, t, \hat{\boldsymbol{c}}) &= D_\theta\left(\boldsymbol{z}_t, t, \begin{bmatrix} \boldsymbol{c}_{\text{sal}} \\ \boldsymbol{c}_{\text{low}} + \Delta\boldsymbol{c}_{\text{low}} \end{bmatrix}\right) \\
&\approx D_\theta\left(\boldsymbol{z}_t, t, \begin{bmatrix} \boldsymbol{c}_{\text{sal}} \\ \boldsymbol{c}_{\text{low}} \end{bmatrix}\right) + \nabla_{\boldsymbol{c}_{\text{low}}} D_\theta\left(\boldsymbol{z}_t, t, \begin{bmatrix} \boldsymbol{c}_{\text{sal}} \\ \boldsymbol{c}_{\text{low}} \end{bmatrix}\right)^\top \Delta\boldsymbol{c}_{\text{low}} \\
&= D_\theta(\boldsymbol{z}_t, t, \boldsymbol{c}) + \sigma_c \nabla_{\boldsymbol{c}_{\text{low}}} D_\theta(\boldsymbol{z}_t, t, \boldsymbol{c})^\top \boldsymbol{n}.
\end{aligned} \tag{24}$$

---

[2]For notational convenience, we omit the explicit $\sigma$ argument in the conditional marginal and write $p_t(\boldsymbol{z}_t \mid \boldsymbol{c}) \equiv p(\boldsymbol{z}_t \mid \boldsymbol{c}; \sigma(t))$.

Compared to CADS, the only change is that the conditioning sensitivity term becomes a subspace-restricted Jacobian–vector product, $\nabla_{\boldsymbol{c}} D_\theta \to \nabla_{\boldsymbol{c}_{\text{low}}} D_\theta$.

Substituting Eq. (24) into the conditional-score Euler update (Eq. (14) with $\hat{\boldsymbol{c}}$) and using Eq. (17), we obtain

$$
\begin{aligned}
\boldsymbol{z}_{t-1} &\approx \boldsymbol{z}_t + \rho_t \frac{D_\theta(\boldsymbol{z}_t, t, \hat{\boldsymbol{c}}) - \boldsymbol{z}_t}{\sigma(t)^2} \\
&\approx \boldsymbol{z}_t + \rho_t \frac{D_\theta(\boldsymbol{z}_t, t, \boldsymbol{c}) - \boldsymbol{z}_t}{\sigma(t)^2} + \rho_t \frac{\sigma_c}{\sigma(t)^2} \nabla_{\boldsymbol{c}_{\text{low}}} D_\theta(\boldsymbol{z}_t, t, \boldsymbol{c})^\top \boldsymbol{n} \\
&= \boldsymbol{z}_t + \rho_t \nabla_{\boldsymbol{z}_t} \log p_t(\boldsymbol{z}_t \mid \boldsymbol{c}) + \frac{\rho_t \sigma_c}{\sigma(t)^2} \nabla_{\boldsymbol{c}_{\text{low}}} D_\theta(\boldsymbol{z}_t, t, \boldsymbol{c})^\top \boldsymbol{n}.
\end{aligned}
\tag{25}
$$

Defining the (time-dependent) noise shaping vector

$$
\boldsymbol{\eta}_t := \frac{\rho_t \sigma_c}{\sigma(t)^2} \nabla_{\boldsymbol{c}_{\text{low}}} D_\theta(\boldsymbol{z}_t, t, \boldsymbol{c}),
\tag{26}
$$

we obtain the following Langevin-like update rule for PADS:

$$
\boldsymbol{z}_{t-1} \approx \boldsymbol{z}_t + \rho_t \nabla_{\boldsymbol{z}_t} \log p_t(\boldsymbol{z}_t \mid \boldsymbol{c}) + \boldsymbol{\eta}_t^\top \boldsymbol{n}.
\tag{27}
$$

Eq. (27) has the same Langevin-style form as CADS, but injects stochasticity only through a selected low-salience subspace of the conditioning, rather than the full conditioning: the first *drift term* follows the conditional score field, while the second *stochastic term* injects Gaussian noise whose covariance is induced by the denoiser's local sensitivity restricted to the low-salience directions. Crucially, because $\boldsymbol{c}_{\text{sal}}$ is kept fixed, the perturbation is confined to the low-salience subspace and thus avoids introducing first-order semantic corruption during sampling.

From this viewpoint, if one sets the low-salience subspace to include the entire conditioning, Eq. (27) recovers CADS; thus, the mask-controlled formulation in Eq. (2) can be interpreted as a generalization of CADS, where CADS corresponds to $M = \mathbf{1}_{L \times D}$ and PADS corresponds to a structured mask that selects only low-salience positions.

To make this restriction concrete, we further assume that the padding-indexed suffix constitutes a low-salience region: Fig. 3 supports this assumption via a noise-sweep analysis, where alignment stays largely unchanged when perturbations are confined to the suffix but drops sharply once the sweep reaches semantically salient coordinates. Consequently, PADS instantiates the low-salience subspace using padding-indexed positions and restricts conditioning noise to this region, providing CADS-style exploration while avoiding first-order semantic corruption and preserving text alignment.

### B.4. Text Conditioning Formats for Applying PADS

For an apples-to-apples comparison across models, we trained SAO as our baseline under the same experimental settings. In our implementation, SAO conditions on a fixed-length text embedding sequence from the text encoder, where the padding entries are set to zero by multiplying the encoder outputs with an attention mask. In contrast, ACE-Step (Gong et al., 2025) and MelodyFlow (Le Lan et al., 2024) are trained with masked attention, and thus do not use an explicit zero-padded embedding for text conditioning.

Table 5. CLAP-f score across models under masked attention and zero-padded text conditioning on the SongDescriber dataset. Bold denotes the strategy used during training.

| Padding-scheme | SAO (Original) | SAO | ACE-Step | MelodyFlow |
|---|---|---|---|---|
| Masked Attention | 0.3597 | 0.3188 | **0.2444** | **0.2783** |
| Zero Padded | 0.3550 | **0.3591** | 0.2425 | 0.2770 |

Table 5 reports how text alignment changes when switching between a zero-padded conditioning format and masked-attention conditioning for each model; the configuration used during training is highlighted in bold (for SAO, the original training configuration is not publicly specified, so no bold marking is provided). Overall, when a model is trained with masked attention, converting it to a zero-padded format yields little change in alignment. Conversely, when a model is trained with zero-padded conditioning, switching to masked attention leads to a small but noticeable change in alignment.

*Table 6.* Performance under different `max_length` values used in the text encoder tokenizer. (a) Performance of SAO on SongDescriber as `max_length` varies. (b) Performance of PADS under different `max_length` values on a dedicated prompt set constructed so that all tokenized prompts have length 128, using $\tau_1 = 0.9$, $\tau_2 = 1.0$, and $s = 0.5$.

(a) SAO on SongDescriber

| max_length | CLAP-f↑ | Vendi ↑ | $KL_{passt}$↓ | $FD_{openl3}$ ↓ |
|---|---|---|---|---|
| **128** | 0.3591 | 19.1763 | 0.5156 | 138.5561 |
| **160** | 0.3539 | 18.6170 | 0.5583 | 139.4426 |
| **192** | 0.3523 | 18.9226 | 0.5673 | 141.5465 |

(b) SAO and PADS on 128-token-length dataset

| max_length | SAO | | SAO+PADS | |
|---|---|---|---|---|
| | CLAP-f↑ | Vendi ↑ | CLAP-f↑ | Vendi ↑ |
| **128** | 0.2791 | 23.1311 | 0.2708 | **24.8903** |
| **160** | 0.2840 | 22.5051 | 0.2755 | **25.6061** |
| **192** | 0.2857 | 22.3300 | 0.2720 | **29.6972** |

Since PADS requires a zero-padded conditioning format, we apply PADS directly to models trained with zero-padded conditioning. For models trained with masked attention, we implement a zero-padded variant; this is supported by the observation above that the alignment change from such conversion is minor.

In the text-to-image domain, Stable Diffusion 1.5 (Rombach et al., 2022) does not use an explicit attention mask in the same way and instead fully utilizes padding positions from the text encoder outputs. Prior studies (Toker et al., 2025; Zhuang et al., 2024; Feng et al., 2023; Li et al., 2024b; Liu et al., 2025b) have shown that, while most information is naturally concentrated in the semantic part of the embedding, padding embeddings can also carry meaningful semantic information, and some work (Ren et al., 2024) further reports that padding may contain summary-like components of the semantic embeddings. Because PADS anneals perturbations in the padding region while allowing the model to leverage whatever information is present there, our formulation is consistent with these observations.

### B.5. Handling Limited Padding in PADS

In Sec. 3.1, we introduced the minimum perturbation budget mechanism, where the last $L_{\min}$ positions are reserved as perturbation slots so that at least $L_{\min}$ locations are always available for noise injection. A potential concern is that, if a prompt is physically long but semantically vague, the resulting perturbation budget may still be too small to provide sufficient diversity, even with the reserved $L_{\min}$ slots. Similarly, for extremely long and detailed prompts with little natural padding, reserving $L_{\min}$ positions could risk perturbing semantically meaningful tokens.

In practice, however, this extreme case is uncommon. Under the standard SAO configuration (`max_length = 128`), tokenized prompt lengths remain well below the available budget, with mean lengths of 29.8 on SongDescriber and 33.4 on MelBench. This indicates that severely limited padding is generally rare in realistic use cases.

More broadly, this issue can be mitigated by increasing `max_length` to restore sufficient padding budget. Table 6 shows that, under the standard SAO setting, varying `max_length` leaves overall generation performance largely unchanged. In addition, on a dedicated prompt set constructed to have token length 128, increasing `max_length` maintains performance at the 128-token setting while providing more padding slots for PADS, thereby increasing diversity without sacrificing text alignment.

## C. MoE-mVAE: Design and Implementation Details

In this section, we first review the fundamentals of prior MoE-mVAE frameworks and then detail our MoE-mVAE architecture inspired by them. We also present results of diffusion models trained in the resulting TAL space learned with the trained MoE-mVAE.

### C.1. Background on Mixture-of-Experts Multimodal VAE

The goal of a multimodal generative model is to (i) stably capture *shared factors* that are common across modalities, while (ii) preserving *modality-specific factors* that are unique to each modality. As a result, two design choices repeatedly become central: *posterior aggregation* (how to combine multiple encoders' posteriors) and *shared–private decomposition* (how to separate shared vs. modality-specific information).

A standard VAE (Kingma & Welling, 2014) introduces an approximate posterior $q_\phi(\boldsymbol{z} \mid \boldsymbol{x})$ to avoid directly optimizing the

intractable marginal likelihood $\log p_\theta(\boldsymbol{x})$, and maximizes the evidence lower bound (ELBO):

$$\mathcal{L}_{\text{VAE}}(\boldsymbol{x}) = \mathbb{E}_{q_\phi(\boldsymbol{z}|\boldsymbol{x})} \left[ \log \frac{p_\theta(\boldsymbol{x}, \boldsymbol{z})}{q_\phi(\boldsymbol{z} \mid \boldsymbol{x})} \right]. \tag{28}$$

This can be equivalently written as the usual reconstruction–regularization form: $\mathbb{E}_{q_\phi(\boldsymbol{z}|\boldsymbol{x})}[\log p_\theta(\boldsymbol{x} \mid \boldsymbol{z})] - \text{KL}(q_\phi(\boldsymbol{z} \mid \boldsymbol{x})\|p(\boldsymbol{z}))$.

MMVAE (Shi et al., 2019) treats each modality encoder as an "expert" and aggregates unimodal posteriors via a Mixture-of-Experts (MoE). Let there be $N$ modalities and denote $\boldsymbol{X} := (\boldsymbol{x}_1, \ldots, \boldsymbol{x}_N)$. Under the conditional independence assumption given a shared latent $\boldsymbol{z}$, the generative model is

$$p_\theta(\boldsymbol{X}, \boldsymbol{z}) = p(\boldsymbol{z}) \prod_{n=1}^N p_{\theta_n}(\boldsymbol{x}_n \mid \boldsymbol{z}). \tag{29}$$

MMVAE defines the joint (aggregated) posterior as the arithmetic average of unimodal experts:

$$q_\Phi(\boldsymbol{z} \mid \boldsymbol{X}) = \frac{1}{N} \sum_{n=1}^N q_{\phi_n}(\boldsymbol{z} \mid \boldsymbol{x}_n). \tag{30}$$

Using this MoE posterior, the MMVAE objective employs *stratified sampling* across modalities (i.e., sampling $\boldsymbol{z}$ from each unimodal expert in turn) and averages the resulting ELBO-like terms:

$$\mathcal{L}_{\text{MMVAE}}(\boldsymbol{x}_{1:N}) = \frac{1}{N} \sum_{n=1}^N \mathbb{E}_{q_{\phi_n}(\boldsymbol{z}|\boldsymbol{x}_n)} \left[ \log \frac{p_\theta(\boldsymbol{X}, \boldsymbol{z})}{q_\Phi(\boldsymbol{z} \mid \boldsymbol{X})} \right]. \tag{31}$$

Expanding the log-ratio using (29) yields a reconstruction term that evaluates *all* modality likelihoods using $\boldsymbol{z} \sim q_{\phi_n}(\boldsymbol{z} \mid \boldsymbol{x}_n)$, plus a regularization term involving the aggregated posterior:

$$\mathbb{E}_{q_{\phi_n}(\boldsymbol{z}|\boldsymbol{x}_n)} \left[ \log \frac{p_\theta(\boldsymbol{X}, \boldsymbol{z})}{q_\Phi(\boldsymbol{z} \mid \boldsymbol{X})} \right] = \underbrace{\mathbb{E}_{q_{\phi_n}(\boldsymbol{z}|\boldsymbol{x}_n)} \left[ \sum_{k=1}^N \log p_{\theta_k}(\boldsymbol{x}_k \mid \boldsymbol{z}) \right]}_{\text{Reconstruction term}} - \underbrace{\mathbb{E}_{q_{\phi_n}(\boldsymbol{z}|\boldsymbol{x}_n)} \left[ \log \frac{q_\Phi(\boldsymbol{z} \mid \boldsymbol{X})}{p(\boldsymbol{z})} \right]}_{\text{Regularization term}}. \tag{32}$$

In (32), the $k = n$ likelihood term corresponds to *self-reconstruction*, while the $k \neq n$ terms correspond to *cross-reconstruction*. Importantly, the regularization does not reduce to a simple $\text{KL}(q_{\phi_n}(\boldsymbol{z} \mid \boldsymbol{x}_n)\|p(\boldsymbol{z}))$ as in a unimodal VAE; instead it includes $\log q_\Phi(\boldsymbol{z} \mid \boldsymbol{X})$, encouraging unimodal posteriors to be consistent with the aggregated MoE posterior.

A persistent limitation of *single shared-latent* formulations is that enforcing strong cross-modal agreement in one latent space can inadvertently dilute modality-specific variation, leading to poor reconstructions for certain modalities and brittle performance under missing-modality settings. The mmJSD framework (Sutter et al., 2020) addresses this issue by explicitly coupling the unimodal posteriors with a multimodal representation through a multimodal Jensen–Shannon divergence regularizer and a dynamic prior construction. Furthermore, in an attempt to better preserve modality-specific information while maintaining a shared semantic space, mmJSD further considers mixture-based multimodal VAEs with a factorized latent space consisting of a shared latent $\boldsymbol{z}$ and modality-specific private latents $\{\boldsymbol{w}_n\}_{n=1}^N$.

Specifically, each modality $\boldsymbol{x}_n$ is generated from both $\boldsymbol{z}$ and its own private code $\boldsymbol{w}_n$, assuming independence between the shared and private latents:

$$p_\theta(\boldsymbol{X}, \boldsymbol{z}, \boldsymbol{w}_{1:N}) = p(\boldsymbol{z}) \prod_{n=1}^N p(\boldsymbol{w}_n) \, p_{\theta_n}(\boldsymbol{x}_n \mid \boldsymbol{z}, \boldsymbol{w}_n). \tag{33}$$

A variational encoder is introduced to approximate posterior inference, and is commonly modeled to mirror these independence assumptions via

$$q_\Phi(\boldsymbol{z}, \boldsymbol{w}_{1:N} \mid \boldsymbol{X}) = q_{\Phi_z}(\boldsymbol{z} \mid \boldsymbol{X}) \prod_{n=1}^N q_{\phi_{\boldsymbol{w}_n}}(\boldsymbol{w}_n \mid \boldsymbol{x}_n), \qquad q_{\Phi_z}(\boldsymbol{z} \mid \boldsymbol{X}) = \frac{1}{N} \sum_{n=1}^N q_{\phi_{\boldsymbol{z}_n}}(\boldsymbol{z} \mid \boldsymbol{x}_n). \tag{34}$$

Achieving scalability in the number of modalities then hinges on how the *joint* shared encoder $q_{\Phi_z}(\boldsymbol{z} \mid \boldsymbol{X})$ is parameterized. In particular, mmJSD discusses that assuming a Mixture-of-Experts (MoE) form for the shared encoder,

$$q_{\Phi_z}(\boldsymbol{z} \mid \boldsymbol{X}) = \frac{1}{N} \sum_{n=1}^{N} q_{\phi_{\boldsymbol{z}_n}}(\boldsymbol{z} \mid \boldsymbol{x}_n), \tag{35}$$

yields a scalable variant of mixture-based multimodal VAEs with shared–private factorization. Under this choice, one obtains an MMVAE-style objective with factorized latent space:

$$\mathcal{L}_{\text{MMVAE}}^{\text{f}}(\boldsymbol{x}_{1:N}) = \frac{1}{N} \sum_{n=1}^{N} \mathbb{E}_{\substack{q_{\phi_{\boldsymbol{z}_n}}(\boldsymbol{z}|\boldsymbol{x}_n) \\ q_{\Phi_{\boldsymbol{w}}}(\boldsymbol{w}_{1:N}|\boldsymbol{X})}} \left[ \log \frac{p_\theta(\boldsymbol{X}, \boldsymbol{z}, \boldsymbol{w}_{1:N})}{q_{\Phi_z}(\boldsymbol{z} \mid \boldsymbol{X}) \, q_{\Phi_{\boldsymbol{w}}}(\boldsymbol{w}_{1:N} \mid \boldsymbol{X})} \right],$$

$$\mathbb{E}_{\substack{q_{\phi_{\boldsymbol{z}_n}}(\boldsymbol{z}|\boldsymbol{x}_n) \\ q_{\Phi_{\boldsymbol{w}}}(\boldsymbol{w}_{1:N}|\boldsymbol{X})}} \left[ \log \frac{p_\theta(\boldsymbol{X}, \boldsymbol{z}, \boldsymbol{w}_{1:N})}{q_{\Phi_z}(\boldsymbol{z} \mid \boldsymbol{X}) \, q_{\Phi_{\boldsymbol{w}}}(\boldsymbol{w}_{1:N} \mid \boldsymbol{X})} \right] = \mathbb{E}_{\substack{q_{\phi_{\boldsymbol{z}_n}}(\boldsymbol{z}|\boldsymbol{x}_n) \\ q_{\phi_{\boldsymbol{w}_n}}(\boldsymbol{w}_n|\boldsymbol{x}_n)}} \left[ \log p_{\theta_n}(\boldsymbol{x}_n \mid \boldsymbol{z}, \boldsymbol{w}_n) \right]$$

$$+ \sum_{\substack{k=1 \\ k \neq n}}^{N} \mathbb{E}_{\substack{q_{\phi_{\boldsymbol{z}_n}}(\boldsymbol{z}|\boldsymbol{x}_n) \\ q_{\phi_{\boldsymbol{w}_k}}(\boldsymbol{w}_k|\boldsymbol{x}_k)}} \left[ \log p_{\theta_k}(\boldsymbol{x}_k \mid \boldsymbol{z}, \boldsymbol{w}_k) \right]$$

$$- \mathbb{E}_{\substack{q_{\phi_{\boldsymbol{z}_n}}(\boldsymbol{z}|\boldsymbol{x}_n) \\ q_{\Phi_{\boldsymbol{w}}}(\boldsymbol{w}_{1:N}|\boldsymbol{X})}} \left[ \log q_{\Phi_z}(\boldsymbol{z} \mid \boldsymbol{X}) \, q_{\Phi_{\boldsymbol{w}}}(\boldsymbol{w}_{1:N} \mid \boldsymbol{X}) \right], \tag{36}$$

where $q_{\Phi_{\boldsymbol{w}}}(\boldsymbol{w}_{1:N} \mid \boldsymbol{X}) = \prod_{n=1}^{N} q_{\phi_{\boldsymbol{w}_n}}(\boldsymbol{w}_n \mid \boldsymbol{x}_n)$ as in (35). Importantly, under this construction, both the self-reconstruction term ($k = n$) and the cross-reconstruction terms ($k \neq n$) are evaluated using the target-modality private posterior $q_{\phi_{\boldsymbol{w}_k}}(\boldsymbol{w}_k \mid \boldsymbol{x}_k)$ when reconstructing $\boldsymbol{x}_k$.

In shared/private models, a known failure mode is that cross-modal reconstruction may exploit a "shortcut" through private latents, i.e., missing-modality reconstructions become overly dependent on modality-specific codes rather than shared semantics. MMVAE+ (Palumbo et al., 2023) targets this failure mode by controlling private information during cross-modal reconstruction: when reconstructing a modality $k \neq n$ from a representation inferred from modality $n$, MMVAE+ does not use $q_{\phi_{\boldsymbol{w}_k}}(\boldsymbol{w}_k \mid \boldsymbol{x}_k)$, but instead resamples $\boldsymbol{w}_k$ from an auxiliary distribution $r_k(\boldsymbol{w}_k)$. Concretely, for each "source" modality $n$, MMVAE+ uses the sampling scheme

$$\boldsymbol{z} \sim q_{\phi_{\boldsymbol{z}_n}}(\boldsymbol{z} \mid \boldsymbol{x}_n), \quad \boldsymbol{w}_n \sim q_{\phi_{\boldsymbol{w}_n}}(\boldsymbol{w}_n \mid \boldsymbol{x}_n), \quad \boldsymbol{w}_k \sim r_k(\boldsymbol{w}_k) \ \forall k \neq n, \tag{37}$$

and evaluates likelihoods for all modalities under $(\boldsymbol{z}, \boldsymbol{w}_{1:N})$. This design explicitly encourages cross-modal generation to depend on the shared latent $\boldsymbol{z}$.

### C.2. Our MoE-mVAE Architecture

Prior works (Shi et al., 2026; Zheng et al., 2026; Liu et al., 2026) have explored improving the latent space used for diffusion modeling, noting that VAE-based latent spaces may lack clear semantic separation or discriminative structure. In this spirit, we adopt a text–audio MoE-style shared–private VAE to construct a text-aware latent space that reflects text–audio semantic correspondence while preserving modality-specific audio information, on which we subsequently train a diffusion model.

Our MoE-mVAE architecture with additional objectives is illustrated in Fig. 12. For each modality, we employ a private encoder $q_{\phi_{\boldsymbol{w}}}$ and a shared encoder $q_{\phi_{\boldsymbol{z}}}$. The latent variables inferred from these encoders are combined to form the decoder inputs, and reconstructions are produced via the audio decoder $p_{\theta_m}$ and the text decoder $p_{\theta_p}$. We denote the private and shared latent variables as follows:

- Modality-specific latent for audio (Private Audio Latent): $\boldsymbol{w}_m$
- Modality-specific latent for text (Private Text Latent): $\boldsymbol{w}_p$
- Shared latent inferred from audio (Shared Audio Latent): $\boldsymbol{z}_m$
- Shared latent inferred from text (Shared Text Latent): $\boldsymbol{z}_p$

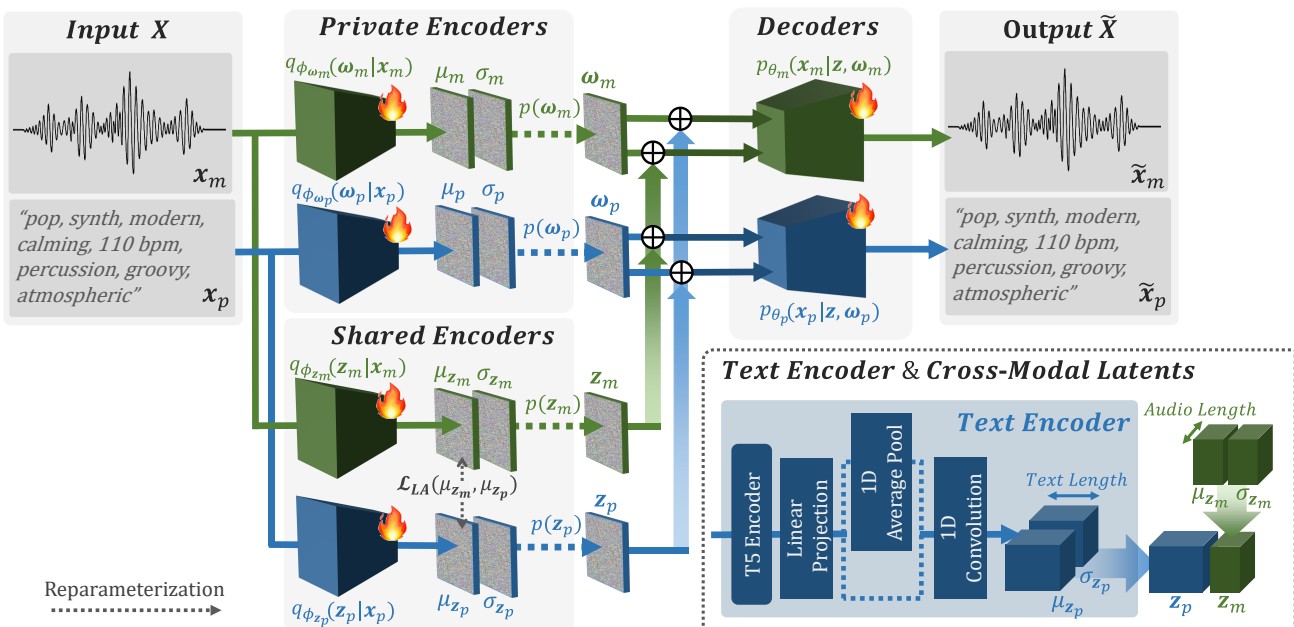

*Figure 12.* Overview of the proposed MoE-mVAE architecture. The model learns private latents $w$ and shared latents $z$ and trains with self/cross reconstructions; the text encoder output is shaped so that, when concatenated with the audio encoder output, the modality fusion occurs along a feature/channel axis that is decoupled from the decoder's time (sequence) axis.

A key component in training our MoE-mVAE is cross-reconstruction. Cross-reconstruction enforces that each modality can be reconstructed by swapping only the shared latent inferred from the other modality while keeping the target-modality private latent fixed, thereby imprinting text–audio semantic correspondence into the latent space. For example, we reconstruct audio by combining the shared latent inferred from text with the private latent inferred from audio, encouraging cross-modal semantics to be reflected in the shared branch of the latent space.

Here, "shared" does not necessarily imply a single identical latent that is universally common across modalities; rather, it refers to the latent branch that is trained to capture factors responsible for cross-modal alignment. Consequently, even when we later train the DM using only the shared text latent, we encourage it to remain aligned with audio-relevant information by adding the LA loss in Eq. (6).

After training our MoE-mVAE, we selectively retain only the modules needed to train the DM on the learned latent space, as depicted in Fig. 6. Specifically, we discard the text decoder and auxiliary pathways used only during VAE training, and keep only the paths necessary to (i) produce a TAL latent from each audio–text pair $(x_m, x_p)$ and (ii) reconstruct audio. Concretely, we obtain $w_m$ via the private audio encoder that captures fine-grained acoustic information from $x_m$, and obtain $z_p$ via the shared text encoder that reflects text semantics aligned with audio. Audio reconstruction is then performed through the audio decoder $p_{\theta_m}$. In summary, TAL is a latent representation designed to jointly contain fine-grained acoustic attributes of audio ($w_m$) and aligned semantic information from the text condition ($z_p$), and the diffusion model learns generation (denoising) in this TAL space.

### C.3. Latent Dimensions and Factorization Details

To perform cross-reconstruction in MoE-mVAE and to construct TAL, the outputs of the text and audio encoders must be combined into a form that can be fed to the decoders. This combination is not merely an implementation detail: it determines the kind of semantic alignment induced by cross-reconstruction, and thus requires careful dimensional design.

If the text latent and audio latent are naively concatenated along the same sequence or time axis, the decoder may inadvertently learn a partitioned dependence in which some temporal segments of the output audio are explained primarily by the audio latent while other segments are explained by the text latent. This unintended segment-wise specialization can weaken the intended training signal of cross-reconstruction—namely, "generating a globally coherent structure of the full audio conditioned on text"—and may encourage a degenerate solution in which only certain time regions depend on text.

*Table 7.* Tensor shapes for input/output of each encoder during VAE and DM training.

| VAE Type | Encoder Type | training VAE | | training DM | |
|---|---|---|---|---|---|
| | | input dim | encoded dim | input dim | encoded dim |
| **Audio-only** | - | $[B, 2, 221184]$ | $[B, 64, 108]$ | $[B, 2, 2097152]$ | $[B, 64, 1024]$ |
| **MoE-mVAE** | **Private Audio $w_m$** | $[B, 2, 221184]$ | $[B, 64, 108]$ | $[B, 2, 2097152]$ | $[B, 64, 1024]$ |
| | **Private Text $w_p$** | $[B, 128, 768]$ | $[B, 32, 108]$ | - | - |
| | **Shared Audio $z_m$** | $[B, 2, 221184]$ | $[B, 32, 108]$ | - | - |
| | **Shared Text $z_p$** | $[B, 128, 768]$ | $[B, 32, 108]$ | $[B, 128, 768]$ | $[B, 32, 1024]$ |

*Table 8.* Tensor shapes of private and shared latent concatenation in MoE-mVAE training.

| | **Shared Audio $z_m$** | **Shared Text $z_p$** |
|---|---|---|
| **Private Audio $w_m$** | $[B, 96(=64+32), 108]$ | $[B, 96(=64+32), 108]$ |
| **Private Text $w_p$** | $[B, 32, 216(=108+108)]$ | $[B, 32, 216(=108+108)]$ |

To avoid this, as shown in Fig. 12, we design cross-reconstruction and TAL construction such that the axis interpreted as time by the decoder is separated from the axis along which modalities are combined. Concretely, we prevent the text and audio latents from being simply appended along the decoder's temporal axis, so that the combination behaves closer to a channel-wise fusion rather than a "time-interval split". Under this design, the dimensions along which $\mu$ and $\sigma$ are formed for reparameterization also naturally differ across modalities, and the latent shapes must be jointly chosen so that the combined representation matches the decoder input format.

A further practical issue is length alignment between text and audio. When the audio encoder/decoder is convolutional, varying input lengths can be handled relatively naturally because the feature length changes flexibly with the input. In contrast, if the text representation relies only on fixed-length pooling or fully-connected transformations, it becomes difficult to expand the text latent to the target latent resolution when the audio length used during VAE training differs from the target audio length used for DM training.

To address this, we design the text encoder output dimensionality with respect to the latent resolution corresponding to the final target audio length, while using 1D pooling during VAE training to enable stable learning even with shorter lengths. As a result, as shown in Fig. 12, the text encoder includes a 1D convolution and linear projection for dimension alignment after the T5 encoder (which provides text embeddings), as well as 1D average pooling for length adjustment. The concrete input shapes and encoded embedding dimensions used in VAE and DM training are summarized in Table 7. For *inputs*, we report the text embeddings as the input to the text encoder (rather than raw token dimensions), and we report the audio embeddings as the input to the audio encoder.

Given these latent designs, the combined latents fed into each decoder are constructed as summarized in Table 8. The audio decoder takes $(w_m, z_m)$ and $(w_m, z_p)$ as inputs, while the text decoder takes $(w_p, z_m)$ and $(w_p, z_p)$ as inputs. Since each decoder must accept a consistent input specification, the corresponding latent combinations are designed to have matching dimensions.

## D. Text-Aware Latent Space Analysis

Although MelBench is not used for training, Fig. 14 suggests that the curated text distribution used for training generally reflects global attributes such as genre, such that genre tends to be relatively well separated even in the text-embedding space alone. In this appendix, we examine what structure the TAL space forms when trained jointly with such text conditioning, quantitatively analyze how well it preserves and organizes global information such as genre, and further provide a brief inspection of generation behavior from the DM trained in TAL.

To study how the TAL space organizes latents in response to genre, we use the MelBench dataset to compare: (i) the latent space obtained from the audio-only VAE encoder of SAO, and (ii) the latent space obtained from the encoders that construct TAL after training the MoE-mVAE. Fig. 5 applies t-SNE to the latent embeddings, with Fig. 5b showing the TAL joint latent $z = [w_m; z_p]$. Consequently, in TAL, genre clusters appear more distinctly separated, with metal and rock forming clearly isolated groups, whereas the audio-only VAE latent space shows greater mixing across genres.

Below, we analyze in more detail how genre-wise distributions form within the TAL latent space and how global attributes

such as genre are reflected.

### D.1. Genre Clustering

We first analyze how clearly the latent representation forms cluster structure at the genre level. For quantitative comparison, we use Silhouette Score (Rousseeuw, 1987), Calinski–Harabasz Index (CHI) (Caliński & Harabasz, 1974), and Davies–Bouldin Index (DBI) (Davies & Bouldin, 1979). The Silhouette Score measures both how compact samples are within their own genre cluster and how well they are separated from the nearest other genre cluster. For sample $i$, let $a_i$ denote the average distance to samples within the same genre cluster, and let $b_i$ denote the minimum (over other genre clusters) of the average distance to that cluster. The Silhouette value is defined as:

$$s_i = \frac{b_i - a_i}{\max(a_i, b_i)}. \tag{38}$$

The overall Silhouette Score is computed as the average of $\{s_i\}$. A positive value indicates that samples are, on average, closer to their own genre cluster than to the nearest other genre cluster, whereas a negative value indicates that samples tend to lie closer to another genre cluster than to their own, suggesting stronger overlap and a higher likelihood of confusion across clusters.

While the Silhouette Score reflects local neighborhood relations at the sample level, CHI and DBI evaluate cluster structure more globally. CHI measures the ratio between inter-cluster dispersion and intra-cluster dispersion. For $N$ samples and $K$ genres, it is defined as:

$$\text{CHI} = \frac{\left( \sum_{k=1}^{K} n_k \|\mu_k - \mu\|^2 \right) / (K - 1)}{\left( \sum_{k=1}^{K} \sum_{x \in C_k} \|x - \mu_k\|^2 \right) / (N - K)}, \tag{39}$$

where $\mu$ is the mean of all latent embeddings, $\mu_k$ is the mean embedding for genre $k$, and $n_k$ is the number of samples in genre $k$. A larger CHI indicates stronger separation between genres and smaller dispersion within each genre, implying clearer genre structure in the latent space.

DBI measures the degree to which each genre cluster overlaps with its most similar other cluster. For $K$ genres, it is defined as:

$$\text{DB} = \frac{1}{K} \sum_{k=1}^{K} \max_{l \neq k} \frac{S_k + S_l}{\|\mu_k - \mu_l\|}, \qquad S_k = \frac{1}{n_k} \sum_{x \in C_k} \|x - \mu_k\|, \tag{40}$$

where $S_k$ denotes the within-genre scatter for genre $k$, defined as the average distance from samples in cluster $k$ to the cluster mean $\mu_k$. A smaller DBI indicates better cluster separation.

Table 9. Clustering performance comparison of latent spaces: audio latent space (SAO VAE) and TAL space (MoE-mVAE).

| Metric | Audio Latent space | TAL space |
|---|---|---|
| Silhouette Score ↑ | -0.0246 | -0.0428 |
| Calinski–Harabasz Index ↑ | 2.7073 | **20.3996** |
| Davies–Bouldin Index ↓ | 25.2582 | **21.5432** |

As reported in Table 9, the Silhouette Score is similar across both latent spaces, indicating comparable preservation of local genre neighborhood structures and reflecting the inherently continuous nature of musical genre boundaries. In contrast, CHI and DBI show meaningful improvements for MoE-mVAE over the audio-only VAE, suggesting that in the TAL latent space, inter-cluster separation increases while intra-cluster compactness is also improved in a more stable manner.

Overall, these results indicate that the TAL-based latent representation on which the diffusion model is trained reflects global semantic structure such as genre more clearly, and provides a representation space that more effectively encodes the global attributes specified by text conditions.

### D.2. Genre Classification

Next, we verify that the latent space does not merely exhibit clustering, but also encodes genre information in a discriminative form. To this end, we train classifiers on the latent embeddings extracted from each model's VAE encoder: Logistic

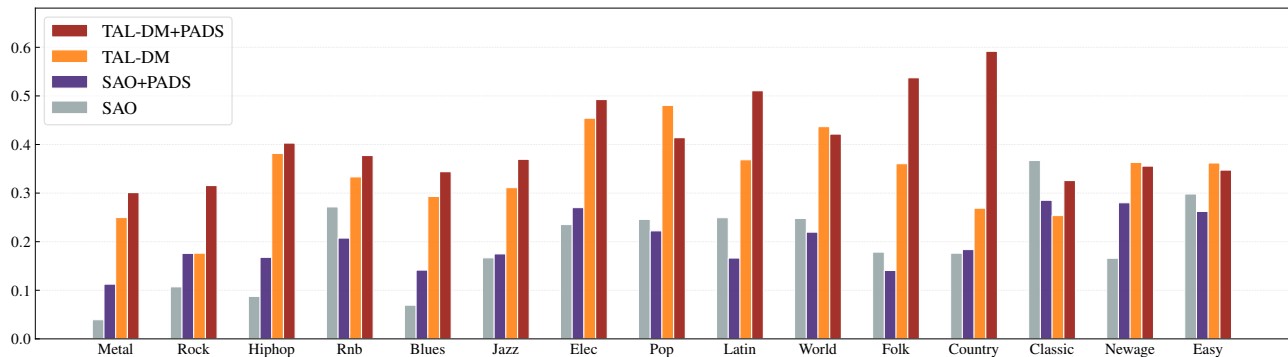

*Figure 13.* $\mathrm{IPR}_{CLAP\text{-}f}$ Recall across all genres on the MelBench dataset ($\tau_1 = 0.8$, $\tau_2 = 1.0$, $s = 0.35$, linear scheduler).

Regression and Random Forest (Breiman, 2001). Logistic Regression is a linear classifier and is used to assess how linearly separable genres are in the latent space. Random Forest is a nonlinear ensemble classifier and is used to evaluate nonlinear genre-discriminative structure captured by the latent representation.

*Table 10.* Downstream classification performance of the audio latent space (SAO VAE) and the TAL space (MoE-mVAE).

| Model | Audio Latent space | TAL space |
|---|---|---|
| Logistic Regression ↑ | 0.0764 | **0.1698** |
| Random Forest ↑ | 0.0778 | **0.1284** |

Table 10 summarizes genre classification accuracy on MelBench. Using the TAL representation yields substantially higher accuracy than the baseline VAE latent for both classifiers. This quantitatively supports that the genre-wise clustering observed in Appendix D.1 reflects learned semantic structure that is genuinely predictive of genre, rather than a purely visual artifact.

### D.3. Generation Behavior of the DM Trained with TAL

To inspect generation behavior of the DM trained in TAL, we evaluate genre-wise coverage on MelBench by measuring how well samples generated from each genre prompt cover the corresponding genre-specific audio reference set. The resulting genre-wise coverage is shown in Fig. 13. (For simplicity, we refer to IPR Recall as "Recall" in the remainder of this section.)

Overall, the TAL-based model exhibits higher coverage across genres than SAO across most genres, suggesting that generation diversity expands to cover a broader set of modes in the reference distribution while maintaining the genre condition. In particular, for genres with salient acoustic and structural characteristics such as Metal, the baseline Recall is low, whereas applying TAL leads to a pronounced increase in Recall. In contrast, for Classic we observe an exception where TAL-DM shows decreased or nearly unchanged Recall compared to SAO. We attribute this to the broad spectrum of sub-modes within Classic (e.g., piano solo, chamber music, symphony/orchestral, and contemporary styles), where the generation distribution may naturally cover diverse modes even under relatively weak conditioning. In this case, strengthening text alignment as in TAL may induce mode bias, where prompt interpretation converges to a particular sub-mode, thereby restricting coverage (Recall) with respect to the reference distribution. Interestingly, under the full pipeline that combines the DM trained in TAL with PADS, Recall for Classic tends to recover, suggesting that perturbations which mitigate overly rigid exploration while maintaining alignment may help reduce genre-dependent discrepancies.

In addition to the analyses above, a broader comparison between the DM trained on the audio-only latent space and the DM trained on TAL is reported in Table 11. While most metrics are comparable, training in TAL alone exhibits a clear tendency toward higher genre coverage. We also observe small improvements in fidelity and in several diversity metrics, indicating that modifying the latent distribution itself can yield measurable gains even without changing the DM architecture.

### D.4. Effect of LA Loss on Latent Structure and Generation

Because TAL is constructed from the trained mVAE using only the text shared latent, rather than both the audio shared latent and the text shared latent, we introduce the LA loss in Eq. (6) to encourage the text shared latent to efficiently capture shared information that is aligned with audio and to better preserve semantically meaningful text information in the resulting

*Table 11.* Comparison between DMs trained on the audio latent and TAL spaces on MelBench and SongDescriber.

| Latent Space | MelBench mRecall↑ | SongDescriber CLAP-f↑ | $KL_{passt}$ ↓ | $FD_{openl3}$ ↓ | Vendi ↑ | IBS ↓ |
|---|---|---|---|---|---|---|
| **Audio** | 0.1937 | 0.3591 | 0.5156 | 138.5561 | 19.1763 | 0.4480 |
| **TAL** | **0.3395** | 0.3546 | 0.4995 | 125.0650 | 20.2426 | 0.4270 |

*Table 12.* Comparison across latent spaces constructed by different VAEs. (a) kNN precision@k measured in each latent space, with a chance level of 0.067. (b) Genre-consistent diversity achieved by diffusion models trained in the corresponding latent spaces.

(a) kNN precision@k in different latent spaces

| Latent Space | LA Loss Applied | kNN precision@k (↑) | | | |
|---|---|---|---|---|---|
| | | k=5 | k=10 | k=20 | k=50 |
| **Audio** | no | 0.066 | 0.068 | 0.067 | 0.066 |
| **TAL** | no | 0.142 | 0.128 | 0.122 | 0.095 |
| **TAL** | yes | **0.243** | **0.211** | **0.189** | **0.164** |

(b) Diffusion-model genre-consistent diversity

| Diffusion Model | LA Loss Applied | mRecall↑ |
|---|---|---|
| **SAO** | no | 0.1937 |
| **TAL-DM** | no | 0.2061 |
| **TAL-DM** | yes | **0.3395** |

latent space.

To empirically examine this effect, Table 12a reports kNN precision@k for the learned Audio Latent and TAL spaces introduced in Fig. 5, including a TAL space trained without LA loss, using genre labels as a proxy for text-level semantics. Since kNN precision@k measures the proportion of same-genre samples within the local neighborhood of each point, it reflects how well semantically related samples form coherent local neighborhoods in the representation space. The original audio-only latent yields values close to the 15-genre chance level of 0.067, suggesting that it captures little text-relevant semantic structure. TAL trained without LA loss already improves neighborhood coherence over the audio-only latent, likely because it still inherits text-related information through the shared–private design. However, TAL trained with LA loss consistently achieves the highest precision, indicating that explicit latent alignment more clearly organizes semantically meaningful text information in the latent space.

This improved latent organization also translates to downstream generation. As shown in Table 12b, removing the LA loss reduces the mRecall of TAL-DM from 0.3395 to 0.2061. Taken together, these results suggest that the LA loss is an important component for aligning the audio and text shared latents, inducing more genre-faithful local neighborhoods in TAL, and improving genre-consistent diversity in the diffusion model trained on top of TAL.

# E. Dataset Details

Recent T2M models condition generation on a wide range of musical attributes, including duration, BPM, instruments, genre, mood, style, and musical composition. In particular, Stable Audio (Stability AI, 2024a) performs conditional generation using structured prompts that explicitly separate attributes such as format, genre/sub-genre, instruments, moods, styles, tempo, and BPM. For example, it uses a delimiter '|' to clearly distinguish fields in prompts such as "Genre: Rock | Instruments: electric guitars | Mood: Moving, Climactic | BPM: 125". In contrast, other T2M models (Evans et al., 2025) often use tag-list prompts, e.g., "dadabots, can't play instruments, pizza hangover, 2021," where it is difficult to explicitly decompose each token (or phrase) into a specific attribute category. Moreover, recent music generation studies also commonly differentiate experimental settings by whether vocals are included and by target duration (e.g., short-form under 10 seconds vs. long-form over 10 seconds).

In this work, we focus on non-vocal (instrumental/BGM) long-form audio of at least 30 seconds, and we standardize text conditioning to tag-list prompts for experimental consistency.

## E.1. Training Data

We construct training data to reproduce the SAO setting as closely as possible, since both the baseline and our model are trained from scratch. The original SAO is reported to be trained on 486,492 audio recordings in total, consisting of 472,618 items from Freesound and 13,874 items from the Free Music Archive (FMA), distributed under CC0, CC BY, or

CC Sampling+ licenses. However, since the original audio files are not publicly released and only metadata (Stability AI, 2024b) is available, reproduction requires re-collecting the source audio based on the released metadata.

In this work, we also perform scratch training based on the same metadata. During collection, some entries could not be downloaded due to missing URLs or missing metadata. After excluding problematic records, we collected audio from the remaining URLs and additionally filtered out files that were excessively short, files smaller than 2KB, and files that caused loading/decoding errors. As a result, we obtained 467,172 Freesound items and 13,122 FMA items for training.

Although the collected training data includes general audio such as sound effects and speech in addition to music, our experiments and analyses are centered on music generation. To better match this objective, we additionally include 17,739 in-house music samples to increase the proportion of music data in training. These in-house samples are used only for training, and we compare models augmented with the in-house music set only against baselines trained under the same augmented-data setting.

### E.2. Evaluation Data

We construct evaluation datasets to assess conditional generation and quality using tag-style prompts for music that is (i) non-vocal (no-singing) and (ii) at least 30 seconds long. We build upon two existing CC BY-SA 4.0 benchmarks, SongDescriber (Manco et al., 2023) and MelBench (Chowdhury et al., 2024), and apply only the minimal preprocessing needed to match our experimental setting (tag-list prompts, non-vocal, long-form).

SongDescriber is used to evaluate overall generation performance. We use 586 no-singing samples provided by Stable Audio Metrics (Stability AI, 2024c). Since the original dataset is caption-centric, we generate tags for the same audio to match the tag-based conditioning used in this paper. Concretely, we first extract a pool of candidate tags using Qwen-Audio (Chu et al., 2023), PANNs (Kong et al., 2020), and SALMONN (Tang et al., 2024), and then filter out words that are irrelevant or weakly related to music to form the final tag set. To incorporate BPM information in the prompt, we compare MADMOM (Böck et al., 2024), Librosa (McFee & librosa development team, 2025), and Essentia (Music Technology Group, 2024), and adopt Essentia, which showed the most stable estimates in preliminary experiments. The estimated BPM is appended to the prompt in a form such as "130 bpm."

MelBench is used for reference-based diversity/coverage evaluation by genre. From 9,311 tracks collected by selecting only downloadable URLs from the provided csv, we construct an evaluation set by balanced sampling of 300 tracks each from 15 genres. Since MelBench contains many vocal tracks, we use instrumental versions obtained by removing vocal components with Ultimate Vocal Remover (Anjok07, 2024) to match our non-vocal setting. We also apply the same tag-generation procedure above to standardize prompts as tag lists.

Fig. 14 visualizes all MelBench prompts by embedding them in the text embedding space and applying t-SNE, with colors indicating genres. Prompts from the same genre exhibit clear clustering and inter-genre separation tendencies, which qualitatively suggests that (i) genre conditions are globally structured already in the text conditioning space, and (ii) although MelBench is only for evaluation (not training), this provides indirect evidence consistent with our design goal in TAL to incorporate text information so that global attributes such as genre are consistently reflected in the latent space.

For reproducibility, we provide the split information and generated tags/metadata for the datasets constructed in this paper along with our code. Since the audio itself is already available via each benchmark and public resource, we do not redistribute audio in order to comply with copyright and licensing requirements.

## F. Evaluation Metrics

Contrastive Language–Audio Pretraining (CLAP) (Wu et al., 2023) is trained on text–audio pairs via contrastive learning so that the text encoder and the audio encoder are aligned in a joint multimodal space. Accordingly, the CLAP score is defined by embedding a given text prompt and a generated (or real) audio sample using the CLAP text/audio encoders and computing their similarity (typically cosine similarity). Thus, a higher CLAP score indicates that the audio is better aligned with the text condition.

Unlike images, however, audio has variable duration across samples. In practice, a fixed target duration is used for model input: shorter samples are padded, and longer samples are cropped. In the Laion CLAP implementation, a fixed 10-second segment is selected via random cropping. Since our evaluation targets music of at least 30 seconds, embeddings vary with the random crop position, and the variance of CLAP scores across repeated measurements can be substantial, reducing

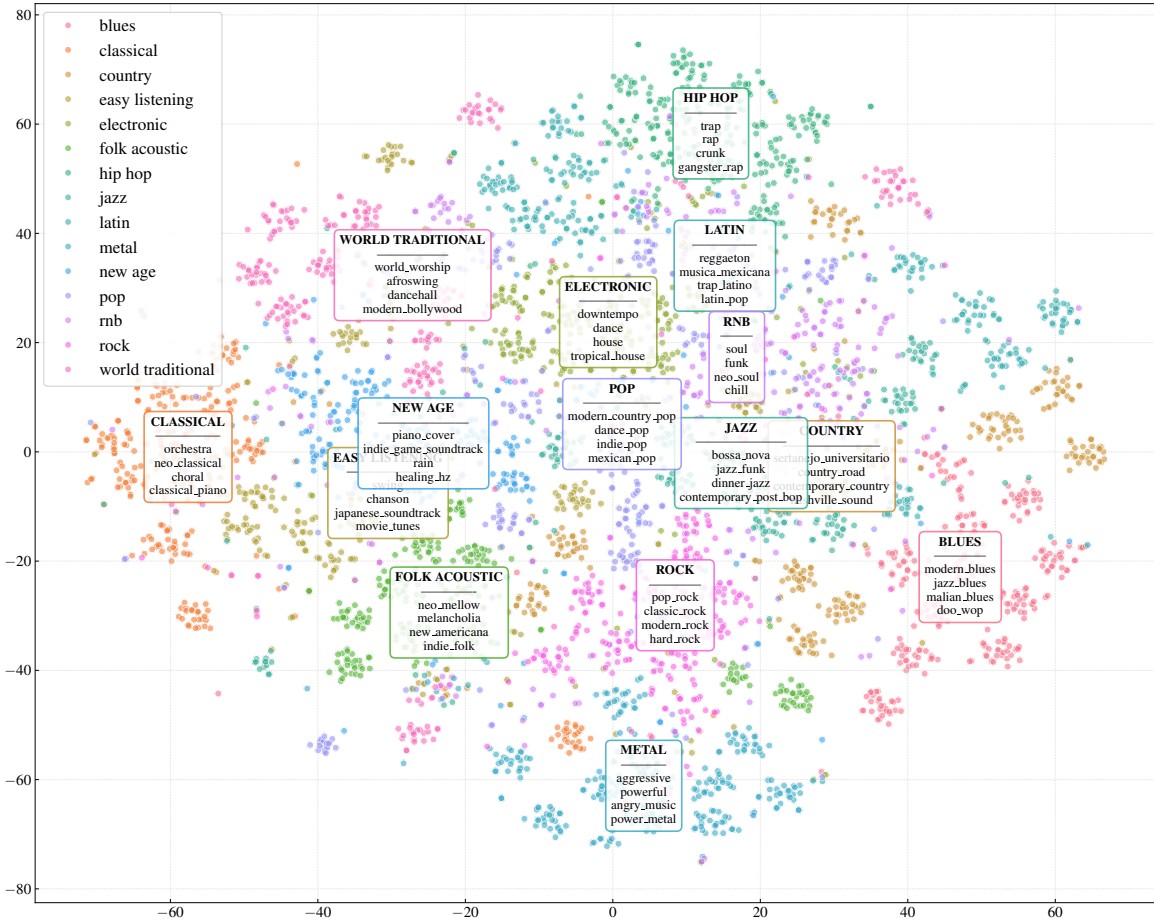

*Figure 14.* t-SNE visualization of genre-specific keywords in the MelBench dataset. TF–IDF features are computed from aspect keywords, and each cluster is annotated with representative genre keywords.

reproducibility.

Moreover, using only a single randomly cropped 10-second segment makes it difficult to evaluate alignment for music that may exhibit different musical form regions (e.g., intro–verse–bridge–outro). A random crop may fail to represent segment-dependent dominant instrumentation, energy, texture, and composition, and even with the same segment length, the embedding and CLAP score can vary significantly with the crop position, further increasing measurement variance.

To mitigate this variance and better measure music, we propose CLAP-f and further describe $\text{IPR}_{CLAP\text{-}f}$ for diversity evaluation based on the resulting embeddings.

### F.1. CLAP-f Score

A straightforward way to reduce the variance of CLAP scores is to perform multiple random crops for the same audio and average the scores. However, this approach requires repeated inference per sample, substantially increasing evaluation time.

To improve both evaluation efficiency and reproducibility, we propose CLAP-fixed (CLAP-f). As illustrated in Fig. 15, CLAP-f extracts CLAP embeddings from three fixed 10-second sub-sequences sampled at 15%, 50%, and 85% positions along the full audio duration (excluding boundary effects implied by the 10-second window). Using all three segment embeddings to compute CLAP-f yields values comparable to multi-crop averaging, while greatly reducing runtime toward near single-pass cost, as shown in Table 13.

Furthermore, CLAP provides both fusion and non-fusion variants. In addition to selecting audio inputs by random cropping, the fusion variant splits the mel-spectrogram into multiple regions, performs random crops within each region, and fuses multiple features to form the final embedding. When applying CLAP-f under the fusion setting, we follow the same fixed-

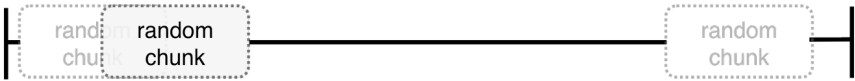

(a) CLAP with random audio segment sampling.

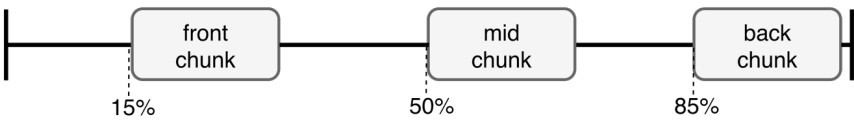

(b) CLAP-f with fixed-position sampling.

*Figure 15.* Comparison of text–audio alignment evaluation mechanisms in CLAP and CLAP-f.

*Table 13.* Comparability between single-run CLAP-f and 10-run averaged CLAP.

| Metric | CLAP (10 runs) | | CLAP-f | |
|---|---|---|---|---|
| | Mean | Range | Mean | Range |
| Result | 0.3132 | [0.2739, 0.3629] | 0.3131 | - |

position principle: we extract audio sub-sequences at 15%, 50%, and 85% and also use mel-spectrogram regions defined by the same fixed-position rule for each sub-sequence. Nevertheless, to focus on music quality, we evaluate text–audio alignment using the music-domain CLAP checkpoint "music_audioset_epoch_15_esc_90.14," which is a non-fusion model.

### F.2. $\text{IPR}_{CLAP\text{-}f}$

Improved Precision and Recall (IPR) (Kynkäänniemi et al., 2019) approximates real and generated data distributions as non-parametric manifolds and separately measures quality and coverage. IPR Precision quantifies the "sample quality/precision" by measuring how many generated samples lie within the real-data manifold, while Recall quantifies the "distribution coverage/diversity" by measuring how many real samples lie within the generated-data manifold.

The original IPR is defined as follows, where the radii for **Data** and **Gen** are determined using the $k$-th nearest neighbor distance:

$$\text{Precision} = \frac{\#\{g \in \mathbf{Gen} \ : \ g \in \mathcal{M}(\mathbf{Data})\}}{\#\{g \in \mathbf{Gen}\}},$$
$$\text{Recall} = \frac{\#\{d \in \mathbf{Data} \ : \ d \in \mathcal{M}(\mathbf{Gen})\}}{\#\{d \in \mathbf{Data}\}}. \tag{41}$$

Here, $\mathcal{M}(\cdot)$ denotes a $k$-NN–based manifold approximation (e.g., a union of balls).

Because embeddings aligned with text conditions are expected to better reflect global (genre/style) information than audio-only embeddings, we use CLAP embeddings for evaluation. Accordingly, to remove variance and non-reproducibility due to random cropping, we use CLAP-f based on fixed segment extraction (15/50/85%). However, CLAP-f produces three sub-sequence embeddings from a single audio sample, and these embeddings tend to be densely clustered in the embedding space. If we define $k$-NN radii using the same rule, then as illustrated in Fig. 16, nearby embeddings derived from the same sample can dominate the $k$-NN neighborhood, artificially shrinking the radius. Since this shrinkage is difficult to eliminate fundamentally due to the structural property that each reference sample contributes three correlated embeddings, we compute IPR independently per sub-sequence position and then average the results. Specifically, for positions $s \in \{15\%, 50\%, 85\%\}$, we define embedding sets $\mathbf{Data}_s$ and $\mathbf{Gen}_s$ and compute Precision/Recall at each position, then average:

$$\text{Precision}_{CLAP\text{-}f} = \frac{1}{3} \sum_{s \in \{15,50,85\}} \frac{\#\{g \in \mathbf{Gen}_s \ : \ g \in \mathcal{M}(\mathbf{Data}_s)\}}{\#\{g \in \mathbf{Gen}_s\}},$$
$$\text{Recall}_{CLAP\text{-}f} = \frac{1}{3} \sum_{s \in \{15,50,85\}} \frac{\#\{d \in \mathbf{Data}_s \ : \ d \in \mathcal{M}(\mathbf{Gen}_s)\}}{\#\{d \in \mathbf{Data}_s\}}. \tag{42}$$

Because $\mathbf{Data}_s$ and $\mathbf{Gen}_s$ are constructed only from embeddings extracted at the same position $s$, this reduces the extent to which nearby embeddings derived from different positions artificially shrink the $k$-NN radius, while ensuring that

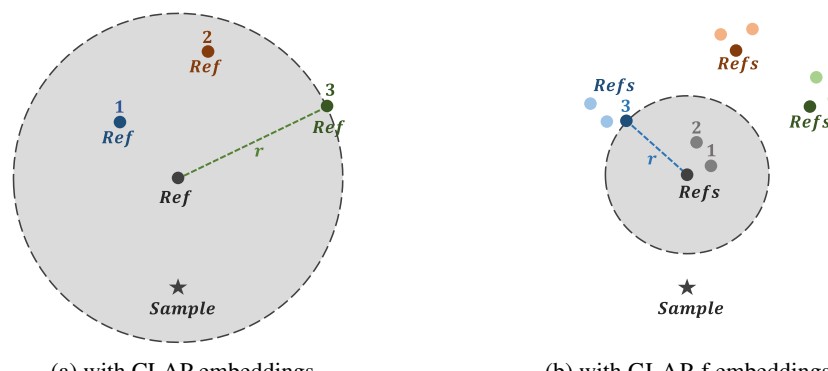

(a) with CLAP embeddings          (b) with CLAP-f embeddings

*Figure 16.* Illustration of IPR with $k = 3$: in (a), the sample is inside the reference manifold with CLAP embeddings; in (b), it is outside with CLAP-f embeddings.

the resulting neighborhood structure reflects position-specific characteristics of the audio, and enables position-matched evaluation that compares structurally similar parts of the music when quantifying diversity.

In this work, we measure diversity across different prompts from two perspectives. First, we evaluate how diverse the model's generated distribution is without reference using Vendi (Friedman & Dieng, 2023). Since Vendi measures the spread of a distribution via an entropy derived from pairwise similarity structure in the embedding space, it is suitable for broadly assessing "free-generation diversity." Second, we measure coverage of a reference distribution under a specific condition (e.g., genre) using Recall.

Accordingly, when evaluating on MelBench as described in Appendix E.2, we construct genre-specific reference embedding sets $\mathbf{Data}^{(c)}$ for each genre and use $\mathrm{IPR}_{CLAP\text{-}f}$ Recall to measure how well the generated distribution covers each genre distribution while remaining diverse.

# G. Additional Experiments and Details

## G.1. Inference Efficiency

To quantify the inference cost of applying PADS, we report the average sampling time per prompt on SongDescriber in Table 14. The DM trained in TAL shows a similar runtime to SAO, and applying CADS or PADS does not substantially change the average sampling time.

*Table 14.* Comparable inference time of SAO and TAL-DM across latent spaces and perturbation methods.

| Models | SAO | | | TAL-DM | | |
|---|---|---|---|---|---|---|
| | Origin | CADS | PADS | Origin | CADS | PADS |
| Inference Time (sec) | 5.1584 | 5.2146 | 5.2194 | 5.1594 | 5.1894 | 5.1955 |

## G.2. CFG-Dependent Behavior

In addition to Fig. 2, we report results under a lower guidance setting, $\omega_{\mathrm{cfg}} = 3.0$, in Fig. 17b. Because lower guidance typically reduces text alignment, the low-guidance setting already yields relatively low alignment and fidelity overall. Under this setting, CADS can still readily collapse in both text–audio alignment and fidelity when $\tau_1$ is not sufficiently small, even with modest injected noise. In contrast, PADS maintains text–audio alignment and fidelity without substantial degradation.

Under low guidance, diversity at matched CLAP-f increases markedly for both CADS and PADS, and PADS achieves slightly higher diversity at the same CLAP-f score. However, the gap is smaller than under high guidance. We attribute this trend to the reduced reliance on the text condition under low guidance, which correspondingly diminishes the influence of the semantic region that PADS aims to preserve.

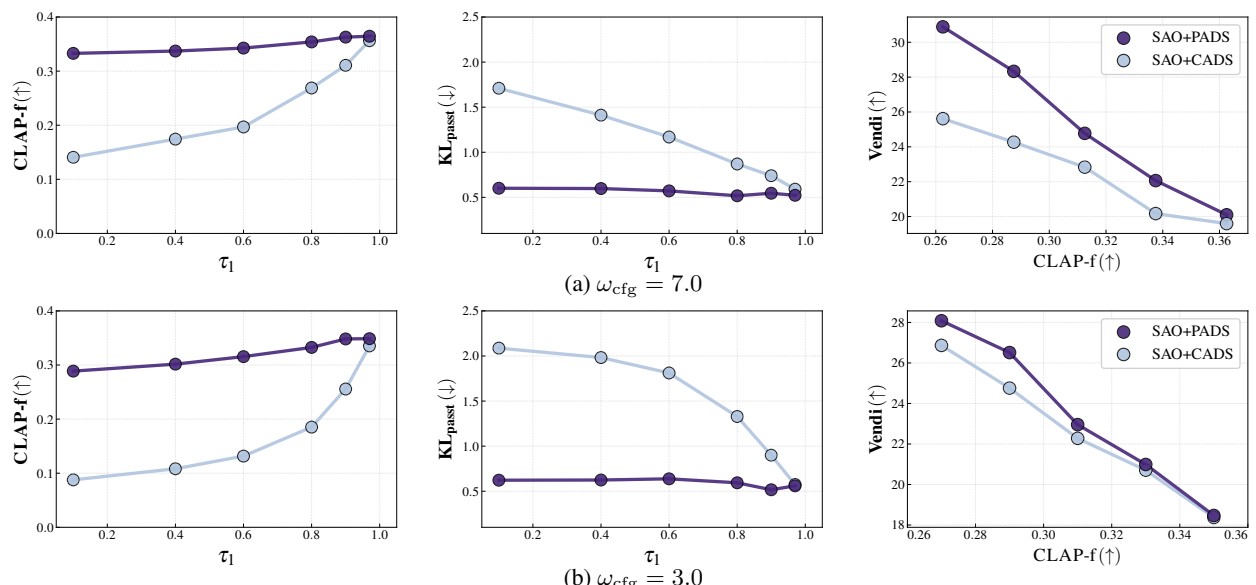

*Figure 17.* Effect of the noise amount on CLAP-f and KLD (left and middle) on the SongDescriber dataset under ($\tau_2 = 1.0$, $s = 0.25$) with a linear scheduler. The rightmost plot shows the CLAP-f–Vendi trade-off.

### G.3. Sensitivity of CADS to Global Information in Text-to-Music

As discussed in Sec. 1, the tendency of CADS to cause rapid collapse in text–audio alignment and audio fidelity as the perturbation strength increases is closely related to diffusion sampling dynamics and the characteristics of the T2M domain. To support this point, Fig. 18 presents the change in text alignment under CADS in both the image and music domains. For the image domain, samples are generated with Stable Diffusion 1.5 (Rombach et al., 2022) on the T2I-CompBench (Huang et al., 2023a) dataset, and changes in text alignment are measured using the CLIP score (Radford et al., 2021). For both domains, we apply CADS with the same parameter setting corresponding to the early-noise regime, and report not only the overall change in text alignment but also the changes associated with global and local text information separately.

In the image domain, the change in global text alignment under CADS is relatively small, whereas the change in local text alignment exhibits a decrease close to that of the overall text alignment. In contrast, in the music domain, the degradation in global text alignment, as measured by CLAP-f, is much more pronounced. This indicates that, when CADS is transferred to T2M, perturbations disrupt global information more readily than in T2I, illustrating why modifications such as PADS are necessary in this domain.

Taken together, these results suggest that CADS induces substantially more sensitive degradation in T2M than in T2I. We interpret this difference as arising from the fact that CADS perturbations act in the early stage of diffusion sampling, where they can more severely damage the global information that is especially important for music generation.

### G.4. Layerwise Sensitivity to Global vs. Local Conditions

We verify through layerwise condition injection experiments that global attributes such as genre/mood/structure and local attributes such as instrumentation/timbre/pitch are not processed in the same way within T2M models.

Fig. 19 follows prior analyses of layerwise prompt roles in UNet-based models (Frenkel et al., 2024; Voynov et al., 2023; Agarwal et al., 2025; Tumanyan et al., 2023). Using a fixed initial noise, we replace the original prompt $p$ (normally injected into all layers) with an alternative prompt $\bar{p}$ that specifies a different attribute, but only at a particular layer (or a set of layers), and evaluate whether the generated output transitions to reflect the attribute specified by $\bar{p}$.

The top two rows show local-attribute experiments, where replacing the prompt at a relatively limited set of layers (the 12th layer) already induces a significant change in local characteristics. In contrast, the bottom two rows show global-attribute experiments, where replacing the prompt at a single layer is insufficient; global attributes tend to transition consistently only when the condition is changed across multiple layers simultaneously. Thus, the assumption that the model can respond differently to global versus local attributes is supported by the observed layerwise sensitivity structure, rather than being an arbitrary conjecture.

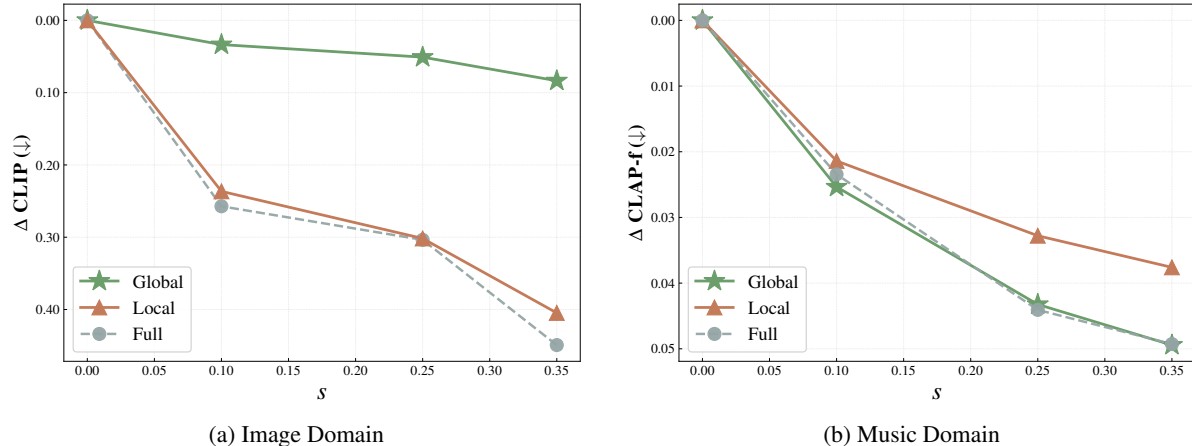

(a) Image Domain                (b) Music Domain

*Figure 18.* (a) CADS-induced changes in CLIP scores for SD under full-prompt, global-only, and local-only prompt settings on T2I-CompBench. (b) CADS-induced changes in CLAP-f scores for SAO under full-prompt, global-only, and local-only prompt settings on SongDescriber. Both (a) and (b) use $\tau_1 = 0.9$ and $\tau_2 = 1.0$, with $s$ varied from 0.1 to 0.35. The corresponding baseline scores are 22.3995 (CLIP) and 0.3591 (CLAP-f).

*Table 15.* Additional results under alternative evaluation settings. For perturbation-based methods, CADS and PADS are reported at matched CLAP-f levels. (**Pert.**: perturbation strategy)

(a) Results on AudioCaps and MusicCaps.

| Architecture | | Evaluation Dataset | | | |
| --- | --- | --- | --- | --- | --- |
| Latent Space | Pert. | AudioCaps | | MusicCaps | |
| | | CLAP-f↑ | Vendi ↑ | CLAP-f↑ | Vendi ↑ |
| Audio | - | 0.239 | 51.488 | 0.294 | 38.922 |
| Audio | **CADS** | 0.213 | 52.672 | 0.267 | 35.231 |
| Audio | **PADS** | 0.212 | **54.088** | 0.268 | **43.556** |

(b) Results on SongDescriber with public SAO* weights.

| Architecture | | Evaluation Dataset | | | |
| --- | --- | --- | --- | --- | --- |
| Latent Space | Pert. | SongDescriber | | | |
| | | CLAP-f↑ | Vendi ↑ | $KL_{passt}$ ↓ | $FD_{openl3}$ ↓ |
| Audio* | - | 0.3550 | 18.7381 | 0.6343 | 140.8122 |
| Audio* | **CADS** | 0.3315 | 22.7478 | 0.7189 | 158.7953 |
| Audio* | **PADS** | 0.3311 | **23.3394** | 0.6985 | 143.7094 |

## G.5. Additional Results and Comparative Evaluations

In this paper, we adopt a controlled evaluation setting—long-form, instrumental, diffusion-based T2M—because T2M evaluation can be strongly confounded by differences between short-form and long-form generation, as well as by whether vocals are included, both of which can introduce substantial variance. For this reason, AudioCaps and MusicCaps are not used as primary benchmarks, since the former is not music-specific and the latter is built from 10-second clips. Nevertheless, since PADS is an inference-time method that can be applied beyond our main evaluation setting, we additionally report results on these datasets in Table 15a as complementary evidence of its broader applicability. The results indicate that our method remains effective on both AudioCaps and MusicCaps under matched CLAP-f.

We further provide comparisons in Table 16 against additional baselines not considered in our main evaluation setting. Specifically, we use `audioldm2-music-665k` for AudioLDM2 (Liu et al., 2024) and `musicgen-melody` for MusicGen (Copet et al., 2023). The public-checkpoint comparison suggests that PADS-TAL compares favorably with these benchmarks, showing strong diversity together with competitive text alignment and fidelity.

In addition, because PADS is an inference-only method, it can be applied directly to the public SAO* weights. Since the public SAO* model is expected to have been trained with masked attention, we apply PADS by slightly extending the low-salience region beyond the simple padding-only region. Table 15b reports results on the official SAO* at matched CLAP-f ($\approx 0.33$). Compared to CADS, PADS improves both diversity and fidelity, confirming that our main conclusions also hold on the publicly available model.

*Table 16.* Comparison of text-to-music generation models, including model family and supported duration, with results reported on SongDescriber. For models that provide multiple variants, we use the variant intended for text-to-music generation, and the classifier-free guidance scale is consistently set to 7.0 across all diffusion models.

| Model | Family | Supported Duration | CLAP-f↑ | Vendi ↑ | $FD_{openl3}$ ↓ |
|---|---|---|---|---|---|
| MusicGen | Autoregressive | 30s | 0.3196 | 14.4348 | 200.1708 |
| Mustango | Diffusion | 10s | 0.2859 | 15.4746 | 306.7720 |
| AudioLDM2 | Diffusion | 10s | 0.2864 | 10.1322 | 269.5180 |
| SAO+CADS | Diffusion | 47s | 0.3195 | 22.5909 | 147.7298 |
| PADS-TAL | Diffusion | 47s | **0.3237** | **26.0746** | **135.5421** |

## G.6. Extending PADS to Image Generation

To examine whether PADS generalizes beyond T2M, we apply it to a T2I application and report qualitative results in Fig. 20. When the baseline model produces low-diversity images for a given prompt, applying CADS can occasionally yield samples with misaligned text conditions or reduced fidelity. In contrast, such failure cases are not observed with PADS in our examples. This supports that PADS can provide diversity more stably while preserving information from the original prompt.

## G.7. Configurations of Diversity Enhancement Methods

Because the diversity enhancement methods used in Fig. 10 induce diversity in different ways, their implementation details and the hyperparameters that govern the diversity–alignment trade-off also differ across methods. Below, we summarize the main implementation choices made when adapting each method to SAO for audio generation, together with the key hyperparameters tuned to obtain the diversity–alignment curves in Fig. 10.

Particle Guidance (Corso et al., 2024) was implemented in its SVGD-like form. In the original implementation, pairwise interactions are weighted by an RBF kernel defined on the Euclidean distance between latent particles. In our implementation, we retain the same particle-interaction structure, but replace the Euclidean-distance-based RBF weighting with cosine-similarity-based weighting, as L2 distance was found to be less informative in the audio latent space. The diversity strength is controlled by a guidance coefficient $\alpha_t$, which is applied only in the high-noise regime.

c-VSG (Askari Hemmat et al., 2024) defines an auto-regressive objective that refers to a memory bank of previously generated samples and a real feature bank, originally constructed in the CLIP feature space of restored samples. However, in the music domain, CLAP features are extracted from mel-spectrograms, and silent regions in music can cause gradient instability. We therefore reformulate the objective using a memory bank built from feature representations extracted from denoised latents, together with a category-wise real feature bank organized by genre. The trade-off curve was obtained by varying the criteria guidance scale $\eta$, which controls the overall magnitude of the normalized guidance gradient, along with the diversity guidance scale $\alpha$, the real-bank weighting scale $\beta$, and the guidance frequency $G_{freq}$.

SPARKE (Jalali et al., 2025) was implemented by following the original framework, while introducing temporal pooling to reduce the computational burden arising from high-dimensional audio features. Specifically, temporal information is aggregated before computing the RKE on the compressed representation. From the perspective of diversity guidance, whereas c-VSG applies a matrix-based entropy score in an auto-regressive manner, SPARKE maps pairwise sample similarity in the latent space through a kernel function and processes it in parallel via batch-wise vectorized operations, yielding improved computational efficiency together with more stable and scalable behavior. The trade-off was controlled by the guidance scale $\eta$ and the guidance frequency $G_{freq}$.

$p$ = "high quality, instrumental, slow tempo, melancholic, nostalgic, piano, romantic, melodic, emotional, sentimental, love song"

$\overline{p}$ = "high quality, instrumental, slow tempo, melancholic, nostalgic, violin, romantic, melodic, emotional, sentimental, love song"

| Replaced | 0 | 1 | 2 | 3 | 4 | 5 | 6 | 7 | 8 | 9 | 10 | 11 | 12 | 13 | 14 | 15 | 16 | 17 | 18 | 19 | 20 | 21 | 22 | 23 |
|---|---|---|---|---|---|---|---|---|---|---|---|---|---|---|---|---|---|---|---|---|---|---|---|---|
| 1 Layer | P | P | P | P | P | P | P | P | P | P | P | P | V | P | P | P | P | P | P | P | P | P | P | P |
| 2 Layers | P | | P | | P | | P | | P | | P | | V | | P | | P | | P | | P | | P | |
| 3 Layers | P | | | P | | | P | | | P | | | V | | | P | | | P | | | P | | |
| 4 Layers | P | | | | P | | | | P | | | | V | | | | P | | | | P | | | |
| 6 Layers | P | | | | | | P | | | | | | V | | | | | | P | | | | | |
| 8 Layers | P | | | | | | | | V | | | | | | | | P | | | | | | | |
| 12 Layers | P | | | | | | | | | | | | V | | | | | | | | | | | |
| 24 Layers | V | | | | | | | | | | | | | | | | | | | | | | | |

$p$ = "An instrumental hopefully positive track where the foreground melody features a classic guitar"

$\overline{p}$ = "An instrumental hopefully positive track where the foreground melody features a piano"

| Replaced | 0 | 1 | 2 | 3 | 4 | 5 | 6 | 7 | 8 | 9 | 10 | 11 | 12 | 13 | 14 | 15 | 16 | 17 | 18 | 19 | 20 | 21 | 22 | 23 |
|---|---|---|---|---|---|---|---|---|---|---|---|---|---|---|---|---|---|---|---|---|---|---|---|---|
| 1 Layer | G | G | G | G | G | G | G | G | G | G | G | G | P | G | G | G | G | G | G | G | G | G | G | G |
| 2 Layers | G | | G | | G | | G | | G | | G | | P | | G | | G | | G | | G | | G | |
| 3 Layers | G | | | G | | | G | | | G | | | P | | | G | | | G | | | G | | |
| 4 Layers | G | | | | G | | | | G | | | | P | | | | G | | | | G | | | |
| 6 Layers | G | | | | | | G | | | | | | P | | | | | | G | | | | | |
| 8 Layers | G | | | | | | | | P | | | | | | | | G | | | | | | | |
| 12 Layers | G | | | | | | | | | | | | P | | | | | | | | | | | |
| 24 Layers | P | | | | | | | | | | | | | | | | | | | | | | | |

$p$ = "fast tempo drums, bass, groovy, urban, electronic, danceable, modern sound, upbeat"

$\overline{p}$ = "fast tempo drums, bass, groovy, urban, jazz, danceable, modern sound, upbeat"

| Replaced | 0 | 1 | 2 | 3 | 4 | 5 | 6 | 7 | 8 | 9 | 10 | 11 | 12 | 13 | 14 | 15 | 16 | 17 | 18 | 19 | 20 | 21 | 22 | 23 |
|---|---|---|---|---|---|---|---|---|---|---|---|---|---|---|---|---|---|---|---|---|---|---|---|---|
| 1 Layer | E | E | E | E | E | E | E | E | E | E | E | E | E | E | E | E | E | E | E | E | E | E | E | E |
| 2 Layers | E | | E | | E | | E | | E | | E | | E | | E | | E | | E | | E | | E | |
| 3 Layers | E | | | E | | | E | | | E | | | E | | | E | | | E | | | E | | |
| 4 Layers | E | | | | E | | | | J | | | | E | | | | E | | | | E | | | |
| 6 Layers | E | | | | | | J | | | | | | E | | | | | | E | | | | | |
| 8 Layers | E | | | | | | | | J | | | | | | | | E | | | | | | | |
| 12 Layers | E | | | | | | | | | | | | J | | | | | | | | | | | |
| 24 Layers | J | | | | | | | | | | | | | | | | | | | | | | | |

$p$ = "This is a perfect sound track for sad ending"

$\overline{p}$ = "This is a perfect sound track for fun happy party"

| Replaced | 0 | 1 | 2 | 3 | 4 | 5 | 6 | 7 | 8 | 9 | 10 | 11 | 12 | 13 | 14 | 15 | 16 | 17 | 18 | 19 | 20 | 21 | 22 | 23 |
|---|---|---|---|---|---|---|---|---|---|---|---|---|---|---|---|---|---|---|---|---|---|---|---|---|
| 1 Layer | S | S | S | S | - | S | S | S | S | S | S | S | S | S | S | S | S | S | S | S | S | S | S | - |
| 2 Layers | S | | S | | - | | S | | S | | S | | S | | S | | S | | S | | S | | S | |
| 3 Layers | S | | | - | | | S | | | S | | | S | | | S | | | S | | | S | | |
| 4 Layers | S | | | | - | | | | - | | | | S | | | | S | | | | S | | | |
| 6 Layers | S | | | | | | - | | | | | | S | | | | | | S | | | | | |
| 8 Layers | S | | | | | | | | - | | | | | | | | S | | | | | | | |
| 12 Layers | - | | | | | | | | | | | | S | | | | | | | | | | | |
| 24 Layers | H | | | | | | | | | | | | | | | | | | | | | | | |

*Figure 19.* Effect of prompt substitution across layers. During generation with the original prompt $p$, we replace the prompt input to the specified layer(s) with $\overline{p}$ and record whether the resulting attribute changes. The top two rows report local attribute changes, while the bottom two rows report global attribute changes. "-" denotes ambiguous outcomes. Labels: P=Piano, V=Violin, G=Guitar, E=Electronic, J=Jazz, S=Sad, H=Happy.

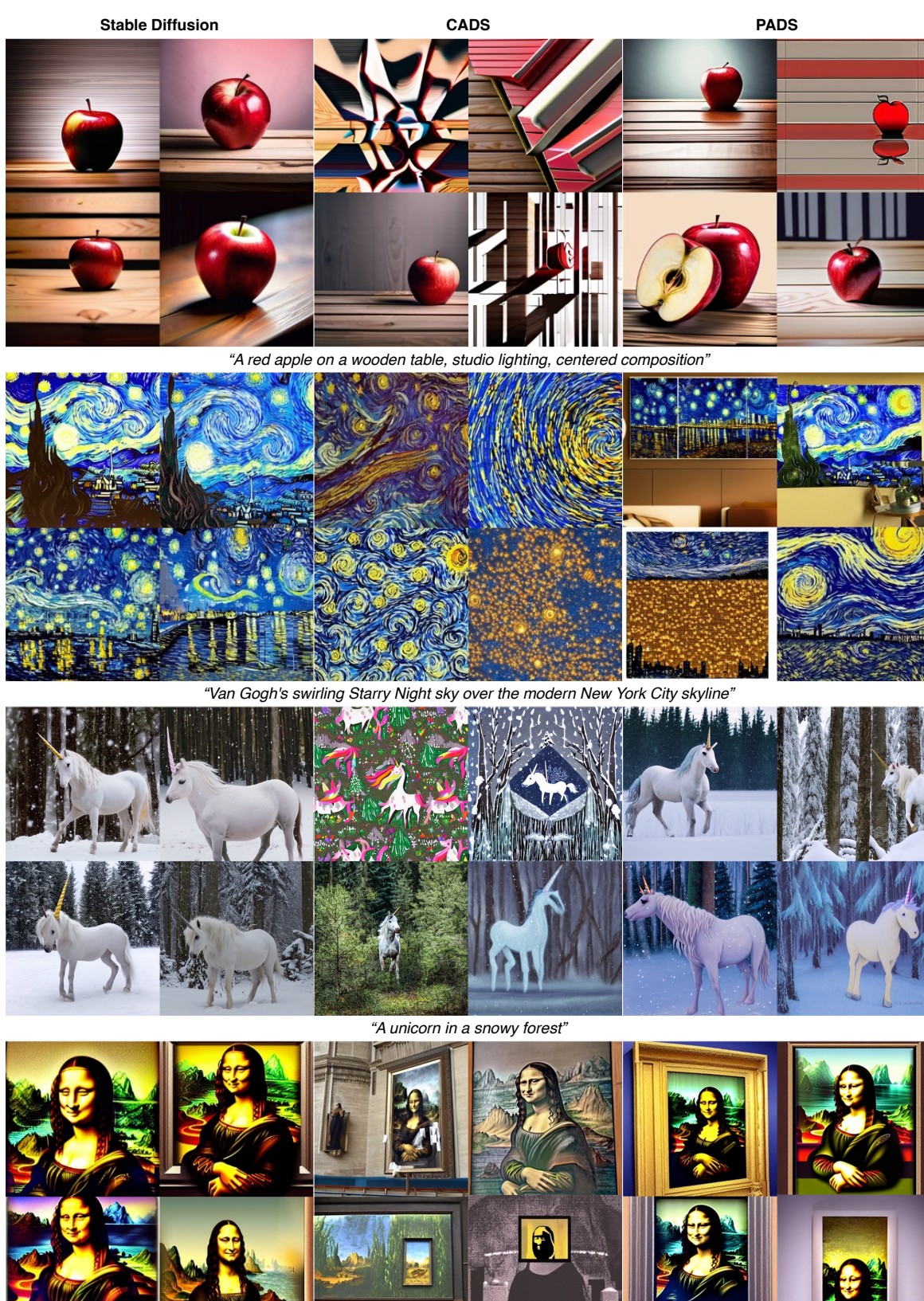

*Figure 20.* All images are generated using Stable Diffusion v1.5 with four different random seeds and fixed sampling hyperparameters (CFG = 7.5, $s = 0.25$, $\tau_1 = 0.6$, $\tau_2 = 0.9$).

*Table 17.* Sampling hyperparameters used across experiments (shared by CADS and PADS). MBD and SDD denote the MelBench and SongDescriber datasets, respectively.

| Target | Dataset | $w_{\mathrm{CFG}}$ | $\tau_1$ | $\tau_2$ | s | Scheduler | $\psi$ |
|---|---|---|---|---|---|---|---|
| **Parameters for Figure 2a** | | | | | | | |
| SAO-CADS | *SDD* | 7 | $[0.1:0.97]$ | 1.0 | 0.25 | linear | 1.0 |
| SAO-PADS | *SDD* | 7 | $[0.1:0.97]$ | 1.0 | 0.25 | linear | - |
| **Parameters for Figure 2b** | | | | | | | |
| SAO-CADS | *SDD* | 7 | $[0.1:0.97]$ | $[0.85:1.0]$ | $[0.01:0.35]$ | linear | 1.0 |
| SAO-PADS | *SDD* | 7 | $[0.1:0.97]$ | $[0.85:1.0]$ | $[0.01:0.35]$ | linear | - |
| **Parameters for Figure 3** | | | | | | | |
| Padding-to-Semantic Sweep | *SDD* | 7 | 0.6 | 0.9 | 0.25 | linear | - |
| **Parameters for Figure 4** | | | | | | | |
| All | *SDD* | 7 | 0.8 | 0.9 | 0.1 | linear | 1.0 (CADS) |
| All | *SDD* | 7 | 0.8 | 0.97 | 0.1 | linear | 1.0 (CADS) |
| All | *SDD* | 7 | 0.8 | 0.97 | 0.25 | linear | 1.0 (CADS) |
| All | *SDD* | 7 | 0.8 | 1.0 | 0.1 | linear | 1.0 (CADS) |
| All | *SDD* | 7 | 0.8 | 1.0 | 0.35 | linear | 1.0 (CADS) |
| All | *SDD* | 7 | 0.9 | 1.0 | 0.1 | linear | 1.0 (CADS) |
| All | *SDD* | 7 | 0.9 | 1.0 | 0.35 | linear | 1.0 (CADS) |
| **Parameters for Figure 7** | | | | | | | |
| SAO-PADS | *SDD* | 7 | 0.1 | 0.85 | 0.25 | linear | - |
| TAL-DM-PADS | *SDD* | 7 | 0.8 | 1.0 | 0.35 | linear | - |
| **Parameters for Figure 8** | | | | | | | |
| SD-CADS | *SDD* | 9 | 0.4 | 1.0 | 0.25 | linear | 1.0 |
| SD-PADS | *SDD* | 9 | 0.4 | 1.0 | 0.25 | linear | - |
| **Parameters for Figure 9** | | | | | | | |
| SAO-PADS | *MBD* | 7 | 0.8 | 1.0 | 0.35 | linear | 1.0 |
| TAL-DM-PADS | *MBD* | 7 | 0.8 | 1.0 | 0.35 | linear | - |
| **Parameters for Figure 10** | | | | | | | |
| SAO-CADS | *SDD* | 7 | $[0.1:0.97]$ | $[0.85:1.0]$ | $[0.01:0.35]$ | linear | 1.0 |
| TAL-DM-PADS | *SDD* | 7 | $[0.1:0.97]$ | $[0.85:1.0]$ | $[0.01:0.5]$ | linear | - |
| **Parameters for Table 1** | | | | | | | |
| ACE-Step-CADS | *SDD* | 7 | 0.8 | 0.97 | 0.1 | linear | 1.0 |
| ACE-Step-PADS | *SDD* | 7 | 0.8 | 0.97 | 0.06 | linear | - |
| MelodyFlow-PADS | *SDD* | 7 | 0.3 | 0.5 | 0.23 | linear | - |
| **Parameters for Table 2** | | | | | | | |
| SAO-CADS | *MBD, SDD* | 7 | 0.8 | 1.0 | 0.10 | linear | 1.0 |
| SAO-PADS | *MBD, SDD* | 7 | 0.1 | 0.85 | 0.25 | linear | - |
| TAL-DM-CADS | *MBD, SDD* | 7 | 0.9 | 1.0 | 0.10 | linear | 1.0 |
| TAL-DM-PADS | *MBD, SDD* | 7 | 0.8 | 1.0 | 0.35 | linear | - |

