# OpenReview forum: "PADS-TAL: Padding-Annealed Diffusion Sampling in Text-Aware Latent Space for Robust and Diverse Text-to-Music Generation"
_ICML.cc/2026/Conference — ICML 2026 regular_

### Official Review · Reviewer_UNuX · 2026-02-22

**Soundness:** 3
**Presentation:** 3
**Significance:** 3
**Originality:** 4
**Overall Recommendation:** 4
**Confidence:** 4

**Summary:**

This paper introduces a unified pipeline to solve the problem of limited diversity and repetitive outputs in Text-to-Music (T2M) diffusion models without sacrificing text-audio alignment or audio fidelity. This work proposes two complementary methods: Padding-Annealed Diffusion Sampling (PADS) and a Text-Aware Latent (TAL) space. PADS is an inference-time strategy that injects exploration noise strictly into the low-salience "padding" subspace of a text prompt, preserving the core semantic embeddings to prevent the semantic drift and fidelity degradation caused by perturbing the entire prompt. To ensure this exploration remains faithful to the requested genre, TAL redesigns the underlying VAE into a multimodal "Mixture-of-Experts" architecture that geometrically aligns the latent space with text-implied genre structures, clustering semantically similar audio-text pairs together. Combined, the unified PADS-TAL pipeline significantly improves the text alignment-diversity trade-off, boosting overall diversity by 15.4% and within-genre diversity by 71.6% over prior full-perturbation methods while maintaining high generation quality.

**Compliance With Llm Reviewing Policy:**

Affirmed.

**Final Justification:**

I appreciate the author's response. I decide to keep my positive score. Thank you.

**Key Questions For Authors:**

1. Why not report the Inception Score (IS) in the results?
2. Why not evaluate text-to-audio on AudioCaps?
3. Why not evaluate text-to-music on MusicCaps?
4. Can the authors add a section elaborating on how Figure 1 is plotted? What does each dot and color represent? How are high- dimensional data points reduced to 2-dimension? How are the data points clustered? How do you draw the diffusion sampling trajectory in latent space into the curves?

**Limitations:**

yes

**Strengths And Weaknesses:**

Strengths:
- Training-Free: PADS is highly practical because it is an inference-time-only strategy. It does not require retraining the underlying diffusion model.
- Instead of relying on a standard, audio-only Variational Autoencoder (VAE), this work introduces the Text-Aware Latent (TAL) space using a Mixture-of-Experts multimodal VAE (MoE-mVAE). This shared-private architecture forces the model's latent space to group semantically similar audio-text pairs together. This structurally solves the problem of "genre mismatch" by ensuring the local neighborhoods of the latent space correctly reflect global text semantics.
- Robust Evaluation Metric: This word noted that standard CLAP scores for long-form audio (30+ seconds) suffer from high variance because they rely on single, random 10-second crops. To fix this, they contribute the CLAP-fixed (CLAP-f) protocol, which extracts embeddings at fixed 15%, 50%, and 85% positions. This provides a more stable, reproducible measurement of text-audio alignment that better accounts for the composition of the whole music track without drastically increasing computational cost.

Weaknesses:
- Lack of broad, system-level comparisons against other state-of-the-art Text-to-Music (T2M) models. The paper completely omits system-level comparisons against other highly prominent, well-known T2M foundation models. For instance, there are no generation quality or diversity comparisons against MusicGen, Mustango, and AudioLDM2.

---

> ### Author Rebuttal · Authors · 2026-03-31
>
> We appreciate you carefully reviewing our paper and raising meaningful questions and constructive suggestions. As mentioned in our response to Reviewer S4df, please kindly refer to **[this anonymous website](https://shorturl.at/Z6Ok5)** for the related materials.
>
> ---
> **W1 : system-level comparisons against other T2M models**
>
> Thank you for suggesting broader system-level comparisons. In this paper, we primarily evaluate our method in a controlled setting—long-form (≥30s), instrumental, diffusion-based T2M—so that diversity and generation quality can be compared under matched conditions. That said, we agree that broader system-level comparisons still provide a useful reference.
>
> **Table F** reports additional evaluations of representative public checkpoints. For fidelity, we focus on FD, as KL can be less reliable when short and long audio are evaluated together due to 10s PaSST windowing and pooling for longer clips. We used audioldm2-music-665k for AudioLDM2 and musicgen-melody for MusicGen, and will include these results in the camera-ready. Overall, the public-checkpoint comparison suggests that PADS-TAL compares favorably to the other benchmarks, with strong diversity as well as competitive text alignment and fidelity.
>
> ---
> **Q1 : Results evaluated with Inception Score (IS)**
>
> We agree that including IS is valuable in our setting, as it jointly reflects diversity and fidelity. The resulting IS evaluations for the generated audio samples from SongDescriber are reported in **Table G**. For perturbation-based methods, we report the results at matched $CLAP$-f, following Table 2 in the paper. As a reference for interpreting IS, we also include the IS values of the ground-truth data in the table caption.
>
> Overall, IS exhibits trends consistent with our main conclusions: under this matched criterion, PADS improves diversity while maintaining generation quality, and TAL-DM+PADS achieves the strongest performance among the compared settings.
>
> We note one practical caveat: IS was computed using the open-source code in [a], which uses a PaSST classifier and requires resampling audio to ≤ 32kHz. Since our default generation setting uses 44.1kHz, we treat IS as a complementary metric and focus primarily on KL, FD, and Vendi.
>
> ---
> **Q2+Q3 : Results generated and evaluated on AudioCaps and MusicCaps**
>
> Thank you for this suggestion. Our primary evaluation focuses on long-form music generation; for this reason, we did not use AudioCaps or MusicCaps as main benchmarks, since AudioCaps is not music-specific and MusicCaps is built from 10-second clips, making it better suited to short-form evaluation than to our 30+ sec setting. That said, we agree that both datasets can still provide useful additional references, especially since SAO itself is trained not only on music but also on more general audio such as sound effects and speech.
>
> For AudioCaps, we used the audio-oriented 630-audioset-fusion-best CLAP model instead of the music-oriented CLAP model used in the paper; for AudioCaps and MusicCaps, we used the corresponding reference embedding sets provided by Stable Audio Metrics [b], following the same experimental protocol as in Table 2 of our paper.
>
> The results are summarized in **Table H** (and will be incorporated into the camera-ready version). Preliminarily, the effect is less pronounced on AudioCaps, though our method still shows a consistent diversity gain.
>
> ---
> **Q4 : Clarification Regarding Fig. 1**
>
> Thank you for pointing this out. We apologize for the lack of clarity. Fig. 1 is a conceptual schematic, not a data-derived plot. Accordingly, the dots, colors, and curve-like trajectories are purely illustrative, and no dimensionality reduction, clustering, or latent-space trajectory tracing on real samples was used to create this figure. We will revise the caption and surrounding text in the camera-ready to make this explicit. For data-based analysis of latent-space structure, we refer to Fig. 5 and the quantitative neighborhood metrics in the paper.
>
> ---
> **Other: Broad comparison with other diversity enhancement methods**
>
> To the best of our knowledge, CADS is the only prior method that explicitly increases diversity by injecting perturbations into the conditioning signal at diffusion sampling time. Nonetheless, to strengthen the empirical context, we additionally include **Figure D**, which compares PADS-TAL against several related sampling-time diversity enhancement methods under a controlled SAO setting. Specifically, we sweep each method’s hyperparameters to trace the text-alignment–diversity trade-off curve, and show that PADS-TAL achieves improved diversity while maintaining stronger prompt alignment in the practically usable regime.
>
> ---
> [a] hkchengrex. av-benchmark. GitHub repository, 2024. https://github.com/hkchengrex/av-benchmark.
>
> [b] Stability AI. stable-audio-metrics. GitHub repository, 2024. https://github.com/Stability-AI/stable-audio-metrics.

---

> > ### Author Rebuttal · Reviewer_UNuX · 2026-04-03
> >
> > - Thank you for the rebuttal. Keeping the score as the application is restricted to instrumental (non-vocal) music.
> > - I am not sure why FreeSound (sound effects) is needed in the training data if the major focus is to generate music. (This question did not pose negative effects to the overall ratings.)

---

> > > ### Author Response · Authors · 2026-04-03
> > >
> > > Thank you for your thoughtful follow-up and for taking the time to read our rebuttal.
> > >
> > > ---
> > > **R1. Clarification on the evaluation setting.**
> > >
> > > Thank you for the comment. Just to clarify one point, we use a controlled evaluation setting—long-form, instrumental, diffusion-based text-to-music—because text-to-music evaluation can be strongly confounded by differences between short-form and long-form generation, as well as by whether vocals are included, both of which can introduce substantial variance. Using a consistent setting enables fair comparisons across methods (Appendix E).
> > >
> > > This choice does not imply that PADS is limited to instrumental music. As shown in **Table H**, PADS also applies to text-to-audio and text-to-music with vocals, and consistently improves over directly applying CADS. We also provide an example on text-to-image generation in Fig. 8 of the paper.
> > >
> > > ---
> > > **R2. Role of FreeSound in training.**
> > >
> > > We appreciate the reviewer’s remark. Just to clarify the training setup, TAL requires training both a text-aware VAE and a downstream DiT from scratch, so a scratch-training setup is necessary to properly demonstrate the method. Accordingly, we used the open-licensed audio sources adopted in SAO-style training, where FreeSound constitutes the majority of the data volume (472,618 items), whereas the explicitly music-focused FMA portion is much smaller (13,874 items).
> > >
> > > In practice, training a robust music generation model from scratch with only the FMA portion is difficult because the amount of music-only data is limited. We therefore included FreeSound, which is not exclusively dedicated to music but does contain music-related audio such as instrumental recordings, loops, and percussive patterns, making scratch training feasible and allowing us to clearly demonstrate the effects of PADS and TAL for long-form music generation.

---

### Official Review · Reviewer_mmkp · 2026-02-23

**Soundness:** 3
**Presentation:** 3
**Significance:** 3
**Originality:** 2
**Overall Recommendation:** 4
**Confidence:** 4

**Summary:**

The paper combines a latent space realignment strategy (TAL) with an elegant inference-time sampling technique (PADS) to solve the "diversity-alignment" trade-off. By focusing perturbations on padding tokens and clustering latents by genre, the method achieves higher musical diversity while maintaining strict consistency with input text prompts.

**Compliance With Llm Reviewing Policy:**

Affirmed.

**Final Justification:**

Thank you for the authors’ response. The reply is largely in line with my understanding of the paper, and I will therefore keep my original score.

**Key Questions For Authors:**

See weaknesses

**Limitations:**

Yes

**Strengths And Weaknesses:**

Strengths:
• The paper explores an interesting task: improving generation diversity while maintaining genre consistency (text-aligned diversity) in text-to-music generation.
• PADS confines noise strictly to padding tokens, providing a zero-cost, training-free balance between semantic preservation and diversity exploration.
• The proposed method effectively combines training-stage latent space reshaping (TAL) with inference-stage sampling (PADS) for a solid, end-to-end solution.
Weaknesses:
• Packaging "CLAP-fixed" (a trivial deterministic cropping heuristic) as one of the three core contributions in the introduction lacks academic depth and appears to be an overstatement.
• The methodological innovation of the paper is somewhat incremental, as the MoE-mVAE architecture is heavily derived from prior works, and the PADS improvement is a straightforward modification of CADS.
• The use of 17,739 private, in-house music tracks for training the models limits the reproducibility of the results and compromises the fairness of the benchmark comparisons.
• The paper fails to evaluate it on the official, publicly available SAO weights, relying entirely on a custom-trained baseline.
• The minimum perturbation budget ($L_{min}$) mechanism forcefully injects noise into the final tokens, which risks destroying the actual semantics of extremely long, detailed prompts that lack natural padding.

---

> ### Author Rebuttal · Authors · 2026-03-31
>
> Thank you for your thorough response. As mentioned in our response to Reviewer S4df, please kindly refer to **[this anonymous website](https://shorturl.at/Z6Ok5)** for the related materials.
>
> ---
> **W1 : Motivation and Positioning of CLAP-f**
>
> We agree that CLAP-f is an evaluation protocol built on top of CLAP via a deterministic cropping rule, and that listing it as a core technical contribution in the Introduction may have been an overstatement. We apologize for the confusion and will revise the contribution list to focus on PADS and TAL, with CLAP-f presented only as an evaluation protocol.
>
> To clarify why we introduced CLAP-f, random 10-second cropping is highly variable for long-form (30+ sec) generation, whereas fixed positions (15/50/85%) provide a more stable and reproducible alignment measure at low additional cost. We therefore mentioned it briefly in the Introduction to motivate our evaluation design, and will re-position it accordingly in the revised manuscript.
>
>
> ---
> **W2 : On the Novelty of PADS and TAL**
>
> We appreciate the reviewer’s concern and clarify what is technically new beyond combining prior components.
>
> **PADS.** We formulate PADS as a mask-controlled generalization of CADS: Eq. (2) defines a unified perturbation family parameterized by $M$, where CADS is recovered with $M=\mathbf{1}$ (an all-ones mask of size L×D) and PADS corresponds to a structured low-salience mask $M=M_{\mathrm{pad}}$.  Appendix B provides a Langevin-like interpretation showing that this choice restricts the stochastic term to the selected subspace ($\nabla_{\boldsymbol{c}}D \rightarrow \nabla_{\boldsymbol{c}_{\mathrm{low}}}D$), while the drift term remains the original conditional-score term. Empirically, Fig. 3 (padding→semantic sweep) supports padding as an alignment-safe instantiation, and Fig. 2b shows higher usable diversity at matched CLAP-f/KL compared to full-conditioning perturbation. In addition, **Figure C** shows that when CADS is transferred to T2M, they disrupt global information more readily than in T2I, illustrating why these modifications are necessary for this domain.
>
> **TAL.** TAL targets a diffusion-trainable text-aware latent space for long-form T2M. While it builds on shared–private principles, making it effective for downstream diffusion required an explicit alignment objective and a latent construction/factorization that decouples the decoder’s temporal axis from the modality-combination axis to avoid segment-wise dependence (Appendix C.3). This improves genre-faithful neighborhood structure (Prec@k) and genre-wise coverage (mRecall).
>
>
>
> ---
> **W3+W4 : private in-house dataset and official SAO public weights**
>
> We agree that reproducibility and fairness are important. However, because TAL requires retraining both a text-aware VAE and a downstream DiT, a direct comparison to the official SAO weights would be confounded by data mismatch: from the released metadata, we could recover 480,294 items, rather than the 486,492 reported in SAO (Appendix E.1). We therefore retrained the SAO baseline on the same reconstructed-data setting used for TAL.
>
> In addition, because the original SAO training set appears to contain relatively limited long-form instrumental music, we added 17,739 in-house music tracks as additional train-only data to better match our target setting and retrained the baseline under the same configuration, while excluding this in-house set entirely from evaluation.
>
> Importantly, PADS is inference-only and can be applied directly to the public SAO* weights. **Table E** reports results on official SAO* at matched CLAP-f (≈0.33): compared to CADS, PADS improves diversity (Vendi 22.748→23.339) and fidelity (KL_passt 0.719→0.699; FD_openl3 158.795→143.709), confirming that our main conclusions hold on the publicly available model. We will clarify these points in the camera-ready and include the official-SAO* inference-only table as part of the main rebuttal evidence.
>
> ---
> **W5 : The Role of $L_{min}$ for Long Prompts**
>
> Thank you for raising this point. Under the standard SAO configuration (`max_length`=128), tokenized prompts are generally far from saturating the fixed-length budget (**Figure B**: mean length is 29.8/33.4 on SongDescriber/MelBench). As a result, cases with severely limited padding are uncommon in practice, and in our reported experiments the perturbation slots remain entirely within padding rather than overwriting semantic tokens.
>
> For unusually long prompts in deployment, we agree that when prompts become both very long and highly detailed, adjusting $L_{min}$ alone may not be sufficient. In such cases, a more appropriate mitigation is to increase `max_length` to restore padding budget. **Table D (b)** shows that, for prompts with token length 128, increasing `max_length` restores enough padding for PADS to be applied. We also discuss the length–specificity issue in our response to Reviewer S4df, Q2, and will make this point explicit in the camera-ready paper.

---

> > ### Author Rebuttal · Reviewer_mmkp · 2026-04-03
> >
> > Thank you for the authors’ response. The reply is largely in line with my understanding of the paper, and I will therefore keep my original score.

---

> > > ### Author Response · Authors · 2026-04-03
> > >
> > > We sincerely thank the reviewer for the positive feedback. We are glad that our clarifications addressed the concerns satisfactorily, and we greatly appreciate the reviewer’s time and consideration.

---

### Official Review · Reviewer_S4df · 2026-03-10

**Soundness:** 3
**Presentation:** 3
**Significance:** 3
**Originality:** 3
**Overall Recommendation:** 4
**Confidence:** 2

**Summary:**

This paper introduces a unified pipeline comprising PADS and TAL to tackle the limited diversity and genre-mismatch problems frequently encountered in Text-to-Music diffusion models.  To prevent the semantic drift caused by existing full-condition perturbation methods like CADS, PADS selectively injects noise only into low-salience padding regions, preserving core prompt semantics while expanding generation exploration. TAL (Text-Aware Latent Space) overcomes the limitations of text-unaware audio VAEs by utilizing a Shared-Private multimodal architecture (MoE-mVAE) to align latent representations with global text semantics, ensuring genre-faithful clustering and generation. To validate the proposed methods, the authors conducted experiments using the SongDescriber and MelBench datasets.

**Compliance With Llm Reviewing Policy:**

Affirmed.

**Final Justification:**

After the rebuttal, my concerns were fully resolved, and I kept my positive score.

**Key Questions For Authors:**

-  The methodology heavily depends on the assumption that the audio and text shared latents are well-aligned due to the LA loss. Are there any specific methods or experiments that could be conducted to empirically verify this alignment?

- PADS relies on the existence of padding tokens to inject exploration noise, assuming that longer prompts naturally require tighter adherence while shorter ones invite varied interpretations. However, if a user provides a prompt that is physically long but semantically vague or lacking in detail, the padding space will be significantly reduced. In such cases, wouldn't PADS fail to work properly and struggle to ensure sufficient diversity?

**Limitations:**

yes

**Strengths And Weaknesses:**

Strength

- The paper is well written and easy to follow.

- The authors tested PADS on an image generation model. This gave clear, visual proof that the method works great.

- PADS can be easily added to various models without making big changes to their systems.

- The structural design and mechanism of the Text-Aware Latent (TAL) space make intuitive sense, and the empirical results clearly demonstrate that it delivers strong generation performance.

Weaknesses

- The paper introduces a Latent Alignment (LA) loss to align the audio shared latent and the text shared latent. However, there is no empirical experiment or quantitative measurement provided to verify whether these two representations are actually well-aligned after training.

- The paper explicitly points out the structural limitations of prior multimodal VAEs. However, to prove the superiority of their redesigned TAL, the authors only perform comparative experiments against the baseline audio-only VAE (SAO). The absence of direct comparisons with other state-of-the-art multimodal VAE architectures leaves the claim that they successfully improved upon multimodal VAE limitations insufficiently validated.

-  In generative model domains, human perceptual quality and naturalness are important. This study relies entirely on automated objective metrics.

---

> ### Author Rebuttal · Authors · 2026-03-31
>
> Thank you very much for taking the time to evaluate our paper and for providing valuable feedback. To offer more detailed results in response to your comments, we have included additional experimental results on **[this anonymous supplement](https://shorturl.at/Z6Ok5)**, and we will incorporate these experiments into the camera-ready version of the paper.
>
> ---
> **W1+Q1: Importance of the LA Loss in TAL training**
>
> We apologize that the paper discussed the motivation for the LA Loss without directly showing its empirical effect.
>
> First, to directly verify whether the audio shared latent and the text shared latent are actually well-aligned after training with the LA Loss, we measured multiple alignment metrics between paired shared latents—cosine similarity and L1/L2 distances—with and without the LA Loss. The corresponding results are provided in **Table A**. As shown there, enabling the LA Loss substantially increases cosine similarity (0.6540 → 0.9185) while sharply decreasing L1/L2 distances (73.7785/65.2240 → 8.1084/6.6522), confirming that LA effectively aligns the two shared representations.
>
> In addition, **Table B** quantifies how this alignment changes the latent neighborhood structure and downstream genre-consistent generation. In the original audio-only latent, kNN precision@k remains near the 15-genre chance level of 0.067 (Prec@5/10/20 ≈ 0.066/0.068/0.067), indicating weak genre-relevant semantic organization. TAL trained without LA already improves neighborhood coherence (Prec@5/10/20 = 0.142/0.128/0.122), and TAL trained with LA further strengthens it (Prec@5/10/20 = 0.243/0.211/0.189). Crucially, this improvement translates to generation: removing LA loss reduces TAL-DM mRecall from 0.3395 to 0.2061. Taken together, these results indicate that the LA Loss is a key ingredient that (i) aligns the audio/text shared latents, (ii) induces more genre-faithful local neighborhoods in TAL, and (iii) improves downstream genre-consistent diversity of the diffusion model trained in TAL.
>
> ---
> **W2: Motivation for our work and the use of TAL**
>
> We apologize for the confusion. Our paper does not claim structural superiority over the multimodal-VAE literature. Rather, our goal is to move beyond the audio-only latent used in SAO-style T2M latent diffusion pipelines and construct a text-aware latent space that better reflects global text semantics (e.g., genre) while preserving audio fidelity. We will revise the Related Work in the camera-ready to avoid any wording that could be read as claiming broad improvements over multimodal VAE architectures.
>
> We agree that head-to-head comparisons across multiple multimodal VAE variants would be informative. However, fully training each alternative VAE and the downstream DiT under our long-form audio setup is computationally expensive (MoE-mVAE ≈ 250 hours on 8×H100; DiT ≈ 320 hours). Given this cost, we prioritized demonstrating TAL’s effect within the target SAO pipeline, supported by our latent-structure and coverage results (e.g., Prec@k and mRecall).
>
> ---
> **W3: Human evaluation for qualitative assessment**
>
> We appreciate you for raising this important point. We agree that, in generative domains such as music and images, human evaluation is important in addition to automated objective metrics. In response, we conducted a human evaluation during the rebuttal period and summarized the results using Mean Opinion Score (MOS). The evaluation was conducted through a manually created survey page, shown in **Figure A**, in a blind setting with 136 participants. Participants were asked to rate quality, diversity, and text alignment on a scale from 1 to 5.
>
> As summarized in **Table C**, the results suggest that PADS and TAL are also effective qualitatively, maintaining high quality while enabling diverse generation.
>
> ---
> **Q2: PADS under long prompts and limited padding**
>
>
> We appreciate this insightful question. In our current evaluation setting, the extreme case where padding becomes severely limited does not arise in practice. Under the standard SAO configuration (`max_length`=128), tokenized prompts remain well below the available budget (**Figure B**: mean = 29.8 on SongDescriber and 33.4 on MelBench), indicating that severely limited padding is generally uncommon in realistic use cases.
>
> More broadly, we agree with the reviewer’s point that prompt length is only a practical proxy for semantic specificity, since even a long but semantically vague prompt may still require high diversity. In such cases, users can (i) remove redundant tokens to shorten the prompt, (ii) increase the perturbation strength within padding (e.g., $s$, $\tau_1$) to encourage more exploration, and/or (iii) increase `max_length` to restore padding budget. **Table D** shows that overall performance remains largely stable across different `max_length` values in (a), and that, for prompts with token length 128, increasing `max_length` restores enough padding for PADS to be applied in (b).

---

> > ### Author Rebuttal · Reviewer_S4df · 2026-04-04
> >
> > Thank you for the authors’ response. My concerns are fully resolved, and I will keep my rating.

---

> > > ### Author Response · Authors · 2026-04-04
> > >
> > > We sincerely appreciate the reviewer’s supportive feedback and thoughtful consideration. We are glad that our explanations helped address the concerns, and we thank the reviewer again for the time and care devoted to evaluating our work.

---

### Decision · Program_Chairs · 2026-04-30

**Decision:**

Accept (regular)

**Comment:**

The paper proposes a coherent pipeline for improving prompt-aligned diversity in diffusion-based text-to-music generation, combining an inference-time method with a text-aware latent space. The empirical results support the central claim that this combination improves the alignment–diversity trade-off and strengthens genre-consistent generation.

The main weaknesses are limited comparison breadth and some overstatement in framing. In particular, the original submission left reasonable questions about latent alignment verification, human evaluation, reproducibility, and comparison against public or broader baselines. The rebuttal materially strengthened the paper by addressing these points, including direct evidence for the LA loss, human evaluation, clarification of the data setting. Reviewers subsequently maintained positive recommendations.

Overall, this is a solid paper with a clear practical contribution. The individual components are not all highly novel, but the overall design is coherent, the gains are meaningful, and the rebuttal addressed the main technical concerns. I support **accept**.